# Structural damage detection and safety assessment method based on machine vision and machine learning

Shengmin Wang[1], Moxiao Li[2,3], Di Le[4*]

1 School of Management, Wuhan Textile University, Wuhan, China, 2 School of Safety Science and Emergency Management, Wuhan University of Technology, Wuhan, China, 3 Hubei Longzhong Laboratory, Xiangyang, China, 4 Wuhan University of Technology, Department of Security, Wuhan, China

* 9394@whut.edu.cn

## Abstract

Structural damage detection and health assessment are crucial for maintaining infrastructure safety and durability. This study presents a novel multi-scale vision-based framework that combines deep learning and machine learning for accurate and interpretable structural safety evaluation. Specifically, we integrate ResNet-50 and SegFormer models to jointly achieve coarse-level damage classification and fine-grained pixel-level segmentation. Seven key damage parameters are quantitatively extracted from high-resolution images—such as crack length, spalling area, and rebar exposure—and serve as interpretable features for safety assessment. A Random Forest (RF) model is developed to establish a nonlinear mapping from these visual features to structural safety levels. Experimental results demonstrate that the RF-based safety assessment model outperforms other traditional machine learning approaches, achieving an accuracy of 87.0%, F1-score of 0.76, and AUC of 0.83, highlighting its strong generalization and classification capabilities. This work offers a comprehensive and generalizable solution for automated structural damage detection and safety evaluation.

## Introduction

Infrastructure development not only embodies the progress of modern civilization but also constitutes a fundamental pillar of socioeconomic advancement. The construction and maintenance of diverse infrastructure systems—including transportation networks, buildings, bridges, and hydraulic structures—are essential to ensuring public well-being and enhancing the overall quality of life. In recent years, the accelerating pace of urbanization and the deepening of economic globalization have significantly increased both the demand for and dependence on infrastructure, in China and worldwide [1–3]. This escalating reliance is primarily driven by factors such as rapid population growth, technological progress, the need for climate-resilient systems,

**Data availability statement:** All relevant data are within the manuscript and its Supporting Information files.

**Funding:** This research was supported by the National Natural Science Foundation of China Youth Project (Grant No. 52209146, Principal Investigator: Moxiao Li) and the Open Fund Project of Hubei Longzhong Laboratory (Grant No. KF-23, Principal Investigator: Moxiao Li). The funders contributed to software development, supervision, validation, and review & editing of the manuscript.

**Competing interests:** The authors declare that they have no conflict of interest.

and the global pursuit of sustainable and interconnected development [4,5]. Nevertheless, the safety and durability of existing infrastructure are increasingly challenged by aging structures, environmental degradation, and the rising frequency of natural hazards such as earthquakes, typhoons, and floods [6].

Under the influence of tensile stress concentration, cyclic loading, and prolonged environmental exposure, structural surfaces tend to deteriorate, typically exhibiting damage such as cracking, concrete spalling, corrosion, and exposed reinforcement [7–11]. Such deterioration not only compromises the load-bearing capacity of structures but may also result in severe safety hazards [12,13]. For example, cracks and concrete spalling directly shorten the service life of roads and bridges, reducing transportation efficiency and increasing safety risks for commuters [14,15]. Likewise, corrosion in reinforced concrete structures significantly weakens the water supply and flood control functions of hydraulic facilities, posing direct threats to public safety and property [16,17]. Consequently, timely and effective detection and maintenance of infrastructure health are essential to safeguarding public safety and promoting sustainable economic development.

Traditional structural damage detection methods primarily rely on manual inspection and conventional measurement tools, such as crack width gauges, ultrasonic testing devices, and ground-penetrating radar [18–21]. Although these approaches provide useful information regarding surface and subsurface damage, they suffer from several well-documented limitations, including low efficiency, high labor intensity, and dependence on inspector expertise. These limitations are particularly pronounced in the assessment of large-scale or geometrically complex structures, where comprehensive and accurate damage evaluation becomes increasingly challenging [22,23]. Furthermore, the subjectivity and variability introduced by human judgment often result in inconsistent and unreliable detection outcomes.

To overcome these limitations, numerous studies have explored automated or semi-automated alternatives for structural damage detection, leveraging advancements in computer vision, image processing, and machine learning. Early efforts focused on conventional image analysis techniques, such as edge detection and texture analysis, which offered limited robustness and generalization in real-world scenarios. More recent approaches have increasingly adopted high-resolution imaging combined with data-driven models, enabling more scalable and objective assessments [24–27]. In particular, deep learning-based methods have shown considerable promise by automatically learning hierarchical and fine-grained features directly from image data, significantly improving the accuracy and efficiency of detecting defects such as cracks, spalling, and corrosion [28–30]. Several studies have investigated the use of deep learning and machine learning models for structural surface damage detection and safety assessment, reporting promising performance metrics. For example, Zhang et al. developed a CNN-based approach for crack detection on pavement surfaces, achieving classification accuracies exceeding 95% [31]. Cha et al. applied a deep convolutional neural network to detect various types of concrete defects in tunnel linings, reaching a precision of 92.3% and a recall of 90.7% [32]. In terms of safety assessment, Chencho et al. employed a random forest-based

approach to quantify element-level structural damage using vibration-derived features, achieving high accuracy and stability in both simulated and experimental scenarios [33]. These studies demonstrate the feasibility of combining surface damage detection with data-driven safety evaluation, although their generalizability across different structural types and image conditions remains a challenge. However, image-based detection approaches predominantly focus on surface-level damage, which may not adequately capture or quantify subsurface deterioration, such as the depth of cracks or the extent of internal spalling. Nevertheless, surface damage often serves as a visible manifestation of underlying structural issues, and thus retains significant value as an indirect indicator for preliminary safety evaluation.

In structural damage detection, machine vision techniques are primarily applied in three domains: object detection [34], damage classification [35], and damage region segmentation [36]. Coarse-grained feature classification networks, such as ResNet [37] and Inception [38], are commonly used for general damage detection. However, fine-grained feature extraction requires more sophisticated network architectures and superior feature extractors [39,40], such as U-Net and Transformer [41,42]. Unlike convolutional neural networks (CNNs), Transformer leverage self-attention mechanisms to capture long-range dependencies in images, thereby enabling more effective global feature extraction [43]. In semantic segmentation tasks, Transformer-based models, such as Swin Transformer [44] and SegFormer [45], have demonstrated outstanding performance in precisely distinguishing objects across multiple scales [46–48]. While Swin Transformer adopts a hierarchical design and window-based self-attention to balance accuracy and efficiency, SegFormer further simplifies the architecture by using lightweight MLP decoders and a pure encoder-decoder structure, making it more suitable for scenarios with limited computational resources. Compared to U-Net++, which relies heavily on CNN-based local operations, SegFormer provides stronger generalization across scales and better global context modeling. This makes it particularly effective for segmenting damage regions with irregular shapes or varying sizes. By integrating image processing technology, the damage information in digital images can be systematically classified and quantified to provide data for structural health monitoring and assessment.

Beyond structural damage detection, structural safety assessment is essential for ensuring the long-term serviceability and reliability of infrastructure. In recent years, data-driven intelligent evaluation methods have gained increasing attention [49–52]. Machine learning-based assessment models can efficiently analyze multi-source data, reducing reliance on complex mathematical modeling and improving evaluation efficiency [53–55]. Methods such as support vector machines (SVM) [56], neural networks (NN) [57], and random forest (RF) [58] have been widely applied in structural health monitoring (SHM) [59]. Among them, the random forest model has particularly robust feature selection capabilities and powerful nonlinear mapping capabilities, which enables it to achieve higher stability and accuracy under limited feature parameters and data samples. By extracting damage features from high-resolution images, machine learning models can map these parameters to structural safety levels, enabling efficient and objective safety assessments.

This study proposes an integrated framework for structural surface damage detection and safety assessment, leveraging computer vision and machine learning techniques. A multi-scale detection approach is employed, wherein ResNet-50 is utilized for image-level classification and SegFormer is applied for fine-grained pixel-level segmentation. Key surface damage types—including material type, number of cracks, total crack length, maximum local crack width, concrete spalling area, crack orientation, and the presence of exposed rebar—are identified and quantitatively characterized through advanced image processing methods. Subsequently, a RF model is constructed to map the extracted damage features to structural safety conditions, enabling an efficient and data-driven assessment of structural integrity. The main contributions of this study are summarized as follows:

(1) A novel multi-scale structural surface damage detection framework is proposed, which integrates the ResNet-50 classification model and the SegFormer segmentation model to effectively handle both coarse- and fine-grained features of structural damage in complex scenes. This cascaded pipeline optimizes computational efficiency by using ResNet-50 as an initial coarse-level filter, ensuring that the heavy Transformer-based segmentation is only applied to regions of interest, thus achieving a superior accuracy-efficiency trade-off compared to single-stage architectures.

(2) A comprehensive image-based damage quantification pipeline is developed, extracting seven key structural damage parameters from high-resolution images using advanced image processing techniques. These parameters serve as interpretable indicators for safety evaluation.

(3) An efficient and generalizable safety assessment model based on the RF algorithm is constructed, establishing a nonlinear mapping between damage features and safety levels. Comparative experiments with other machine learning models validate the superiority of the proposed approach.

The remainder of this paper is organized as follows: Methods introduces the methods for extracting damage features. Experimentation and verification presents experimental validation and quantification of characteristic parameters. Structural safety assessment describes the structural safety assessment methodology and results, followed by the conclusion in Conclusion.

## Methods

### Damage detection methods

In practical engineering, structural surface damage detection based on machine vision still faces several technical challenges, primarily in the following three aspects:(a) Limited detection accuracy, making it difficult to precisely identify and quantify different types of damage; (b) Insufficient detection efficiency, which fails to meet the real-time monitoring requirements of large-scale engineering structures; and (c) The difficulty in balancing detection accuracy and efficiency effectively. Researchers have proposed various surface damage detection methods for different detection tasks, such as damage classification and damage segmentation, as discussed in the previous section. Fundamentally, these methods solve the problem of damage detection at different levels of granularity. At the image block level, image blocks are classified as containing or not containing damage. Damage segmentation is performed at the pixel level. Each pixel in the image domain is divided into a damage region and a background region. For example, the segmentation model needs to perform 10,000 classification tasks, while the image classification model only needs one classification task, when detecting a 100×100 pixels image. From the perspective of computational efficiency, the image patch-based classification method is more suitable for large-scale engineering applications where real-time detection is crucial.

To enhance both the efficiency and accuracy of structural surface damage detection, this study proposes a cascaded detection approach that integrates damage classification and segmentation. The technical workflow of this approach is illustrated in Fig 1.

The input of the detection model is a high-resolution original image of the structure. The ResNet-50 classifier is used to identify whether the image contains concrete spalling information from a macroscopic level. The surface images of the structure obtained by machine vision (such as drones and robots) are divided into two categories. For the image set of concrete spalling, a ResNet-50 classifier is used to identify whether there is exposed steel in the spalling area. Then, a SegFormer model is used to segment the concrete spalling area at the pixel level. Compared with concrete spalling with obvious pixel features, cracks are not easy to identify due to their slender morphology. Therefore, the identification of cracks requires a more local and detailed method. For the image set that does not contain concrete spalling, the sliding window model is used to segment the original image containing damage information into multiple small-sized image blocks. Subsequently, the image classifier based on ResNet-50 is used to classify each image block to determine whether it contains cracks. The image blocks that are judged to have cracks are segmented using the SegFormer model. Thus, the morphological details of the cracks are obtained.

### Classification model

To enhance both the accuracy and efficiency of surface damage detection, the original image is first divided into small-scale image blocks, each measuring 112×112 pixels. ResNet-50 is employed as the backbone network to construct the damage classification model. The network structure is shown in Table 1.

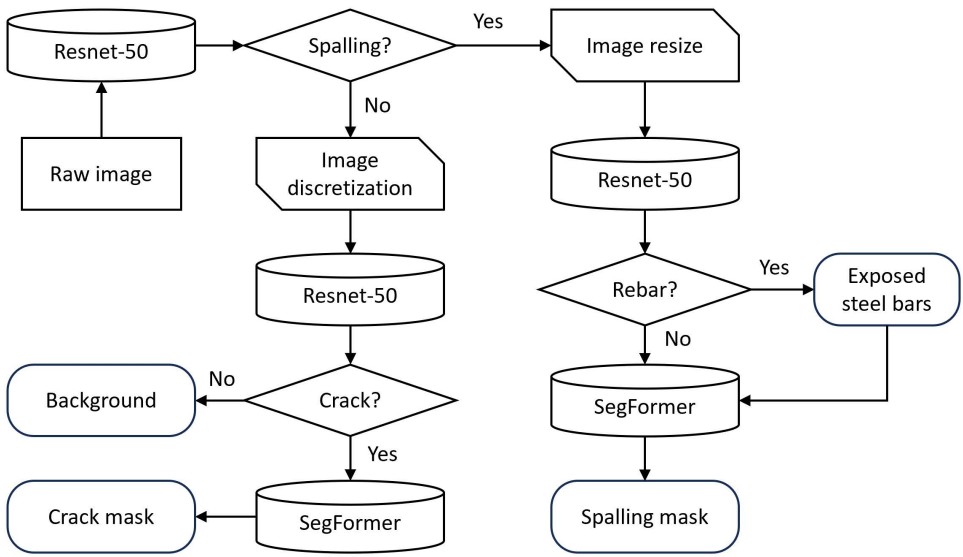

**Fig 1. Technical flow chart of damage detection.**

**Table 1. Network structure table of the classifier.**

| Layer Index | Layer Type | Kernel Size / Parameters | Output Channels | Activation Function | Output Dimensions (H×W×C) |
|---|---|---|---|---|---|
| 1 | Input | – | 3 | – | 112×112×3 |
| 2 | Convolutional | 3×3, stride=1, padding=1 | 64 | ReLU | 112×112×64 |
| 3 | Residual Block ×1 | Two 3×3 convolutions | 64 | ReLU | 112×112×64 |
| 4 | Residual Block ×1 | Two 3×3 convolutions | 64 | ReLU | 112×112×64 |
| 5 | Residual Block ×1 | Two 3×3 convolutions | 64 | ReLU | 112×112×64 |
| 6 | Global Avg Pooling | – | – | – | 1×1×64 |
| 7 | Fully Connected | 64→128 | 128 | ReLU | 1×1×128 |
| 8 | Fully Connected | 128→2 | 2 | Softmax | 1×1×2 |

The first convolutional layer employs a 3×3 convolutional kernel with a depth of 64 and utilizes the ReLU activation function to introduce non-linearity. The core of the ResNet-50 network consists of residual blocks, each comprising two 3×3 convolutional layers and an identity mapping. By leveraging residual connections, the model effectively alleviates the vanishing gradient problem, thereby preserving both depth and complexity. Three residual blocks are utilized to extract feature representations from image patches. Following the residual blocks, a fully connected layer with 128 neurons is incorporated, where the ReLU activation function facilitates nonlinear feature transformation and integration. Finally, the output layer consists of two neurons corresponding to the damage and non-damage classifications, with probabilities computed using the Softmax activation function.

The classifier is trained using the cross-entropy loss function, defined as follows:

$$\mathcal{L}_C = -\frac{1}{N} \sum_{i=1}^{N} \left[ y_i \log(p_i) + (1 - y_i)\log(1 - p_i) \right]$$

(1)

where $N$ represents the number of samples, $y_i$ denotes the sample label, and $p_i$ is the predicted probability by the model.

The Adam optimizer was used for model training. The learning rate, batch size, and number of iterations were set to 0.001, 32, and 200, respectively. To ensure the generalization ability of the model, the dataset was split into a training set and a validation set in a ratio of 8:2.

## Segmentation model

Transformer models are renowned for their ability to handle long-range dependencies, making them particularly suitable for capturing global features in images. Unlike traditional CNN, Transformer models employ self-attention mechanisms to effectively model long-distance dependencies between pixels, thereby achieving superior performance in image segmentation tasks. As a lightweight segmentation network within the Transformer family, SegFormer has been widely adopted for object segmentation in specific scene images. The network structure is shown in Fig 2.

The SegFormer model partitions the input image into multiple small patches, where self-attention calculations are performed within local regions. Initially, the pre-classified crack image is fed into the segmentation model. Through a hierarchical structure, these local features are progressively integrated to obtain global information of the entire image. At the core of the SegFormer architecture, each patch undergoes processing by multiple layers of Transformer encoders, which consist of multi-head self-attention (MSA) mechanisms and feedforward neural networks (FFNs) [60]. This enables the model to extract high-level features from image patches effectively. The extracted features are then passed to the decoder stage, where each pixel is classified as either a crack pixel or a background pixel.

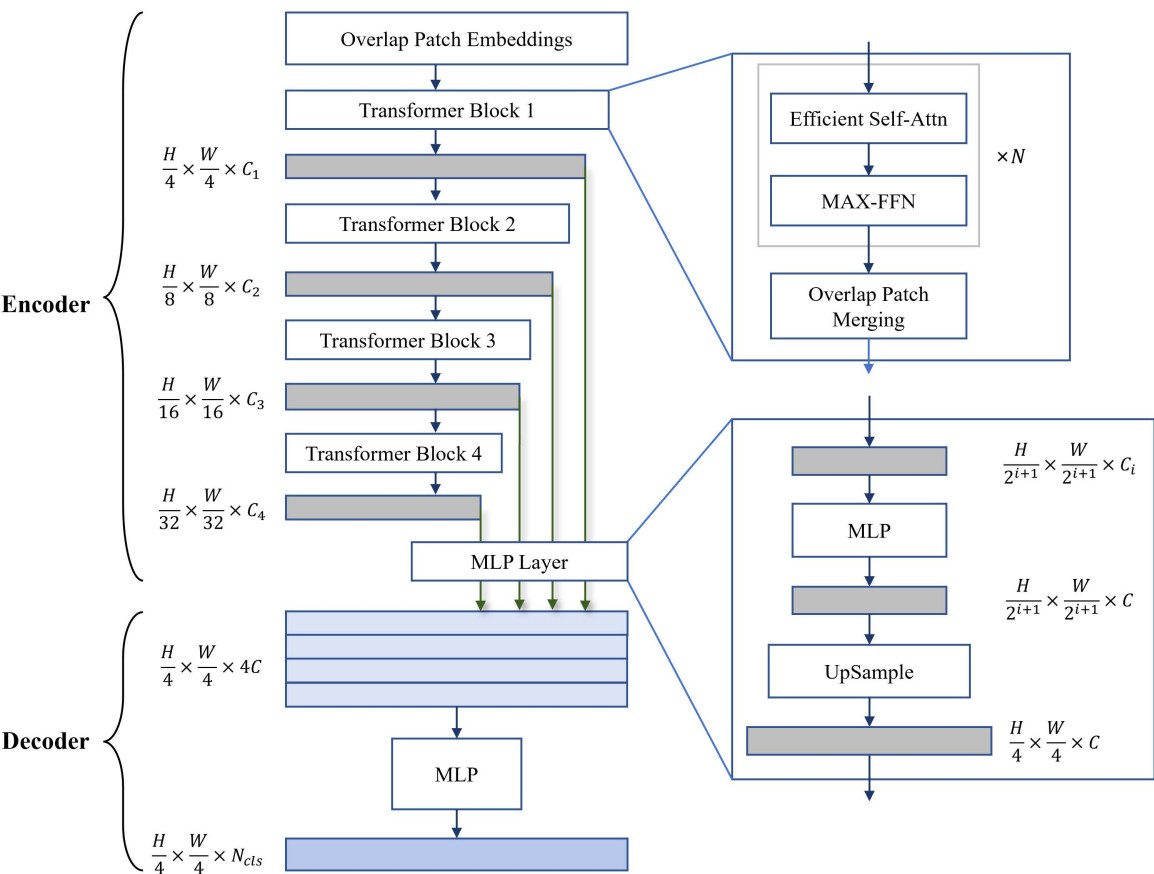

**Fig 2. The network structure of SegFormer [45].**

To optimize the crack segmentation model, a combined loss function comprising cross-entropy loss [61] and Dice loss [62] is utilized. The formulations are as follows:

$$\mathcal{L}_{CE} = -\frac{1}{N}\sum_{i=1}^{N}\sum_{c=1}^{C} y_{ic}\log(p_{ic})$$

(2)

$$\mathcal{L}_{Dice} = 1 - \frac{2\sum_{i=1}^{N} y_i p_i}{\sum_{i=1}^{N} y_i + \sum_{i=1}^{N} p_i}$$

(3)

$$\mathcal{L}_S = \alpha\mathcal{L}_{CE} + (1-\alpha)\mathcal{L}_{Dice}$$

(4)

where $N$ represents the total number of pixels, $C$ is the number of classes, $y_{ic}$ denotes the ground truth label of pixel $i$ for class $C$, and $p_{ic}$ represents the predicted probability of pixel $i$ belonging to class $C$. Similarly, $y_i$ is the actual label of pixel $i$, and $p_i$ is the predicted probability of pixel $i$ being classified as the positive class. The coefficient $\alpha$ is empirically set to 0.5 in this study to equally balance the contributions of cross-entropy and Dice losses, achieving a trade-off between pixel-level accuracy and shape-level segmentation quality.

## Damage quantification methods

Based on the damage detection method, this section further uses image processing technology to extract geometric parameters of different damage types for quantitative analysis.

To quantify the cracks, the orthogonal skeletonization method is applied to extract the skeleton of the segmented crack regions, ensuring that the skeleton centerline accurately reflects the crack morphology [63]. The total crack length is then computed using the region growing algorithm [64], while the shortest path algorithm is utilized to eliminate crack branches, ensuring accurate measurements. Crack width information is obtained by performing multi-segment linear scanning perpendicular to the crack skeleton, providing an accurate estimation of the maximum local width. Additionally, hough transform is applied to determine the primary orientation angle of the crack, thereby evaluating its propagation trend [65]. For concrete spalling, the edges of the segmented damage regions are extracted. The contour approximation algorithm is then employed to compute the spalling area.

## Evaluation method of crack detection results

To evaluate the performance of the crack detection methods, the accuracy serves as an intuitive and effective reference index for the crack classification model. For the crack segmentation model, precision, intersection over union (IoU), recall, and F1-score are usually used as evaluation metrics. These segmentation evaluation metrics are defined based on the confusion matrix, where:

True Positive (TP): The number of correctly predicted positive samples.

True Negative (TN): The number of correctly predicted negative samples.

False Positive (FP): The number of incorrectly predicted positive samples.

False Negative (FN): The number of incorrectly predicted negative samples.

For binary classification tasks, accuracy measures the model's ability to correctly classify images into the appropriate category, defined as:

$$\text{Accuracy} = \frac{TP + TN}{TP + TN + FP + FN}$$

(5)

Precision represents the ratio of correctly classified crack pixels to the total number of pixels classified as cracks, defined as:

$$\text{Precision} = \frac{TP}{TP + FP} \tag{6}$$

IoU quantifies the overlap between the predicted crack region and the ground truth region, defined as:

$$\text{IoU} = \frac{TP}{TP + FP + FN} \tag{7}$$

Recall measures the proportion of actual crack pixels correctly identified by the model, defined as:

$$\text{Recall} = \frac{TP}{TP + FN} \tag{8}$$

F1-score is the harmonic mean of precision and recall, defined as:

$$\text{F1} = 2 \times \frac{Precision \times Recall}{Precision + Recall} \tag{9}$$

**Crack quantitative measurement method**

This paper uses the orthogonal skeletonization method to identify the contour and skeleton of the crack and calculate the crack length. The method is an image processing technique used to extract the skeleton of an object in a binary image. The skeleton refers to the central axis of the object after gradual refinement. It can retain the topological structure of the object and remove redundant information, thereby simplifying complex shapes. The specific implementation steps are as follows:

(1) Contour extraction

Canny edge detection is used to extract crack contours, which define the boundary between crack regions and the background, accurately describing the crack shape.

(2) Skeleton extraction

Thinning operations are performed on the binary image to gradually shrink the crack area and retain its center line [66]. Then, the thinning result is used to generate the skeleton line of the crack. The distance between the two intersection points perpendicular to the skeleton line and the crack edge is the crack width.

(3) Crack length calculation

The skeleton line consists of a series of connected pixels. The length of the crack is calculated by accumulating the distances of these pixels. For pixels in the horizontal and vertical directions, the distance between adjacent pixels is 1. For pixels in the diagonal direction, the distance is $\sqrt{2}$. Let the sequence of pixels on the skeleton line be $(x_i, y_i)$, and the calculation formula for the skeleton length $L$ is:

$$L = \sum_{i=1}^{N-1} d((x_i, y_i), (x_{i+1}, y_{i+1})) \tag{10}$$

where,

$$d((x_i, y_i), (x_{i+1}, y_{i+1})) = \begin{cases} 1, & \text{if } |x_i - x_{i+1}| + |y_i - y_{i+1}| = 1 \\ \sqrt{2}, & \text{if } |x_i - x_{i+1}| = 1 \cap |y_i - y_{i+1}| = 1 \end{cases} \tag{11}$$

Image segmentation-based method is employed to precisely identify and quantify the area of concrete spalling. The objective of concrete spalling detection is to extract the damage boundary and compute the damaged pixel area, providing an estimation of the severity of the spalling. The specific implementation steps are as follows:

(1) Contour extraction and edge detection

Canny edge detection is used to extract the boundary contours of the spalling region. The canny operator, which computes image gradients, effectively extracts closed boundaries of the damage region, thereby improving the stability of recognition. Given an original image $I(x, y)$, edge detection is formulated as:

$$G(x, y) = \sqrt{(I_x^2 + I_y^2)}$$

(12)

where $I_x^2$ and $I_y^2$ represent the horizontal and vertical gradients, respectively, and $G(x, y)$ denotes the gradient magnitude image. By setting appropriate high and low thresholds, weak edges are filtered out, retaining only the primary contours of the concrete spalling region.

(2) Area calculation

Closed contour regions are extracted. And the pixel area is computed as:

$$A = \sum_{(x,y) \in \Omega} 1$$

(13)

where $\Omega$ represents the detected spalling region.

## Experimentation and verification

In this section, computational models are developed using the Python programming language and the PyTorch deep learning framework, with GPU acceleration employed to enhance training and inference efficiency [67, 68]. The proposed detection method is validated using images of concrete spalling and exposed rebar collected from real-world engineering structures, thereby demonstrating its practical applicability. To ensure consistency and accuracy throughout the workflow, model training, testing, and inference are all conducted within the same hardware environment. The computational platform consists of an Intel® Xeon® W-2123 processor (3.60 GHz), 64 GB of RAM, and an NVIDIA GeForce RTX 2080 Ti GPU.

## Data processing

**Generate dataset.** Crack images of different structural styles and material types were collected through three sources: public datasets, online retrieval, and on-site photography. The dataset includes images of reinforced concrete pavement cracks, cracks on concrete structural surfaces, and cracks on steel structural surfaces. All images required manual annotation to generate labeled datasets for training the classification and segmentation models, regardless of whether the images were obtained from public datasets or captured by researchers.

Typically, the image resolution of engineering structures varies depending on the capturing device. Additionally, due to computational limitations of hardware, common semantic segmentation tasks require input images to be divided into small-sized image patches before being fed into the crack segmentation model. For classification tasks, image blocks were manually categorized into background images and crack images based on the presence of cracks. Some samples are shown in Fig 3.

A total of 200 high-resolution crack images were collected from roads, buildings and bridges. Additionally, some publicly available crack detection datasets (including SDNET2018, CRACK500, DeepCrack, CrackForest, and Road Crack Dataset) were incorporated to expand the datasets [31,-69–73]. After image preprocessing, the dataset was generated as summarized in Table 2.

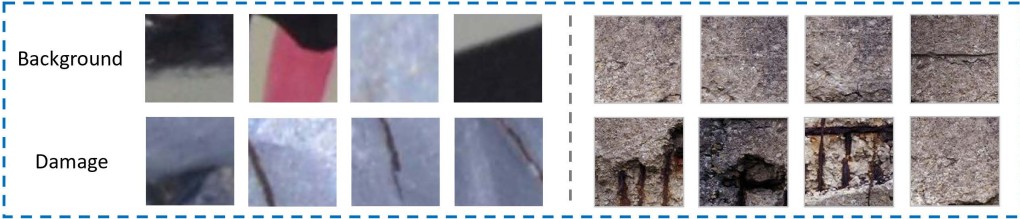

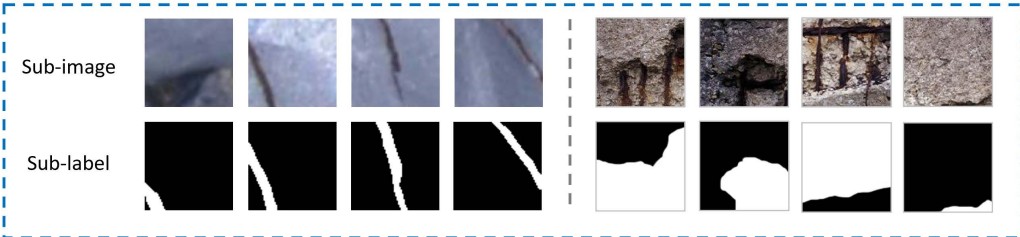

**Fig 3. Sample dataset.**

**Table 2. Composition of the dataset.**

| Model | Train datasets | | Test datasets | |
|---|---|---|---|---|
| **Concrete spalling classifier** | **Background** | **Target** | **Background** | **Target** |
| | 2000 | 2000 | 500 | 500 |
| | 2000 | 2000 | 500 | 500 |
| **Crack classifier** | 4000 | 4000 | 1000 | 1000 |
| | 4000 | 4000 | 1000 | 1000 |
| **Concrete spalling segmentation** | **Image** | **Label** | **Image** | **Label** |
| | 4000 | 4000 | 1000 | 1000 |
| | 4000 | 4000 | 1000 | 1000 |
| **Crack segmentation** | 2000 | 2000 | 500 | 500 |
| | 2000 | 2000 | 500 | 500 |

## Data augmentation

To enhance the robustness and generalization ability of the model, data augmentation techniques were introduced during the data preprocessing stage [74]. Data augmentation applies a series of transformations to the original images, generating diverse training samples to prevent overfitting and improve the model's adaptability to different scenarios.

Data augmentation was performed by applying various transformation operations to the original images, generating new training samples and effectively expanding the dataset. The augmentation techniques employed in this study include rotation, flipping, and random cropping.

Each image underwent a random rotation within a range of ±15 degrees, as defined by:

$$I'(x, y) = I(xcos\theta - ysin\theta, xsin\theta + ycos\theta) \tag{14}$$

Images were randomly flipped along horizontal and vertical axes to simulate cracks appearing in different orientations, as defined by:

$$I'(x, y) = \begin{cases} I(W - x, y); & \text{if horizontally flipped} \\ I(x, H - y); & \text{if vertically flipped} \end{cases} \tag{15}$$

Images were randomly cropped within 90% to 110% of its original size to simulate cracks at different scales, as defined by:

$$I'(x, y) = I(x + \Delta x, y + \Delta y) \tag{16}$$

Through data augmentation techniques, the dataset was tripled, significantly increasing the diversity and richness of the training samples. These enhanced images enable the model to better learn crack features, thereby improving its accuracy and robustness in practical applications.

### Image patch classification results

A classification model was developed based on the ResNet-50 network to distinguish background and damages in structural surface images. During the training process, the best model was saved by monitoring the loss value on the validation set. As illustrated in Fig 4, the concrete spalling image classifier began to converge after 10 iterations, with the loss values of the training set and test set alternating while stabilizing around 0.05. The accuracy was used as the evaluation metric for model performance. After 7 iterations, the accuracy of both the training and validation sets had already exceeded 0.96. As the number of iterations increased, the accuracy of the concrete spalling classifier further improved, approaching 0.99.

For the crack classification model, the loss function curve began to converge after 15 iterations, while the accuracy curve surpassed 0.95 after 10 iterations. Compared to concrete spalling images, crack patterns are finer and more intricate, making feature extraction more challenging for the ResNet-50. However, the training results indicate that the classifier was able to maintain high accuracy for binary classification tasks at the image level.

All models achieved their peak performance within the 50 training epochs, indicating that further iterations were unnecessary.

To evaluate the performance of the classification model, original structural images that were not included in training were selected for testing. The test results are shown in Fig 5. From the prediction results, the original image was divided into multiple 112×112 image patches. Image patches classified as containing cracks were retained at their corresponding positions in the original image, whereas image patches classified as not containing cracks were replaced with solid color fills.

Based on the binarized crack image, the crack orientation is fitted, as indicated by the red lines in the figure. Rebar exposure typically coexists with concrete spalling; however, not all spalled concrete areas necessarily contain exposed rebar. As shown in the fourth row of Fig 5, image patches containing concrete spalling are covered by white pixels, while exposed rebar is represented by gray pixels.

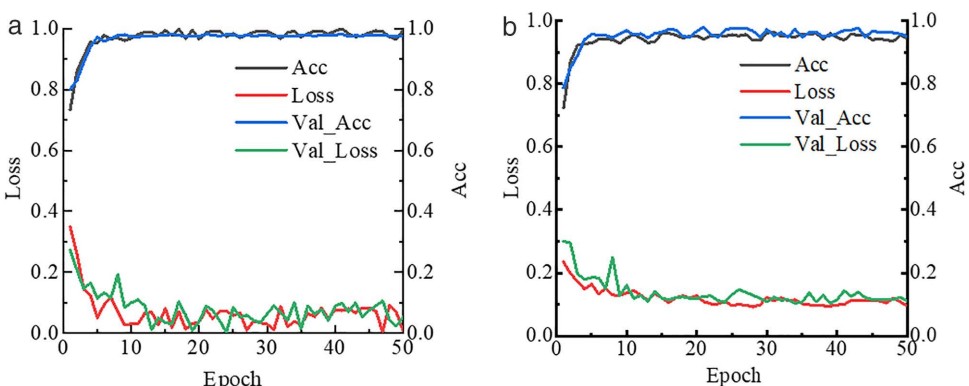

**Fig 4. The training curves of classification model.** (a) Concrete spalling classifier. (b) Crack classifier.

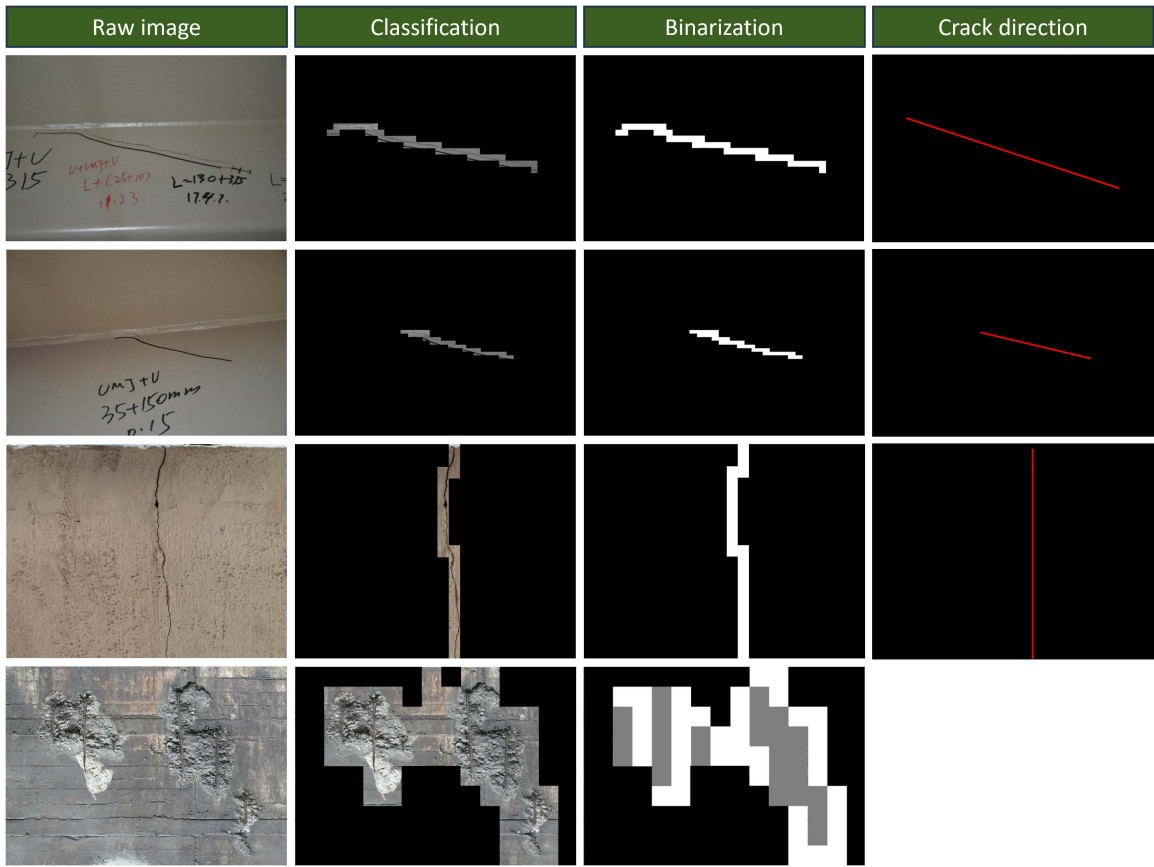

| Raw image | Classification | Binarization | Crack direction |
|---|---|---|---|

**Fig 5. Example of classification results of the original image.**

When extracting damage features, crack orientation is classified into three categories: "horizontal," "vertical" and "inclined." Specifically, if the inclination angle of the red line is less than 20 degrees or greater than 160 degrees, the crack is considered horizontal. If the inclination angle falls between 70 and 110 degrees, the crack is classified as vertical. All other inclination angles correspond to the inclined category. Rebar exposure is categorized into two classes: the presence or absence of exposed rebar.

## Damage segmentation results

A crack segmentation model is trained using the SegFormer. The training curve of the damage segmentation model is illustrated in Fig 6. From the trend of the loss function curve, it can be observed that the concrete spalling segmentation model starts to converge gradually from the 46 iterations. Ultimately, the loss function stabilizes at approximately 0.038 on the training set, which is lower than the 0.073 observed on the validation set. The IoU for the concrete spalling segmentation model reaches 0.935 on the training set and 0.879 on the validation set. The convergence trend of the crack segmentation model is generally consistent with that of the concrete spalling segmentation model. However, the IoU of the crack segmentation model is slightly lower than that of the concrete spalling segmentation model. The IoU for the crack segmentation model reaches 0.882 on the training set and 0.818 on the validation set.

To evaluate the performance of the segmentation model, original structural images that were not included in the training process were selected for testing. The test results are shown in Fig 7. As observed from the prediction results,

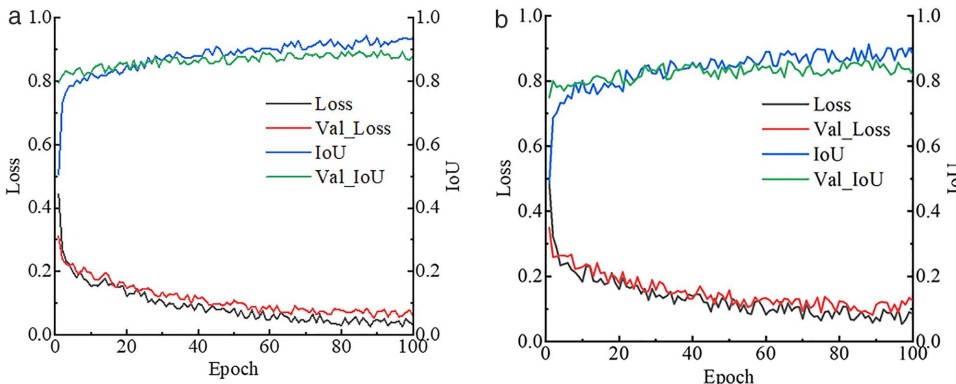

**Fig 6. The training curves of segmentation model.** (a) Concrete spalling segmentation. (b) Crack segmentation.

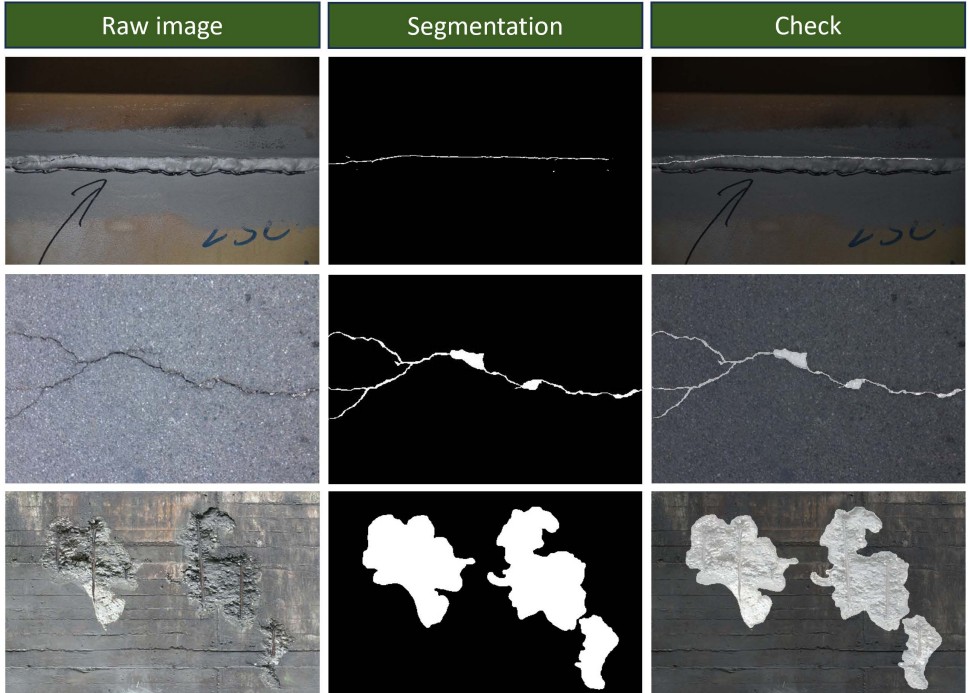

**Fig 7. Example of segmentation results.**

the crack patterns in the original images were accurately segmented. The predicted binary crack images effectively capture the actual crack characteristics, including spatial location, morphology, length, and width, preserving crucial structural details.

## Crack measurement results

The segmented binary images of cracks undergo further processing. By applying the Canny edge detection algorithm and the contour tracing algorithm, the extracted crack contours achieve high clarity. The edge detection method accurately

captures the crack boundaries, while the contour tracing algorithm effectively reconstructs the complete crack morphology, as illustrated in Fig 8.

The orthogonal skeleton line method is employed to accurately measure crack length by identifying contour lines and skeleton lines in the binarized crack image. The refined skeleton lines precisely represent the central path of the crack while eliminating redundant branches and noise points, ensuring smoothness and continuity. After processing multiple crack images, the extracted crack contour lines clearly depict the morphology and propagation direction of the cracks, as illustrated in Fig 9. The measured crack widths and skeleton lengths are statistically presented on the right side of Fig 8. Given the elongated shape of cracks, this study records the top three maximum crack width values for each detected crack. To validate these measurements, we compared the results with manual measurements using digital calipers on 50 samples, achieving a Mean Absolute Error (MAE) of 0.05 mm for width and 2.4 mm for length, which meets the requirements of routine structural inspections.

## Structural safety assessment

By utilizing the visual image processing methods proposed in previous sections, this study successfully extracted multiple key structural surface damage features, including crack length, width, orientation, spalling area, and presence of exposed rebar. These damage parameters not only describe the degree of structural surface degradation but also provide

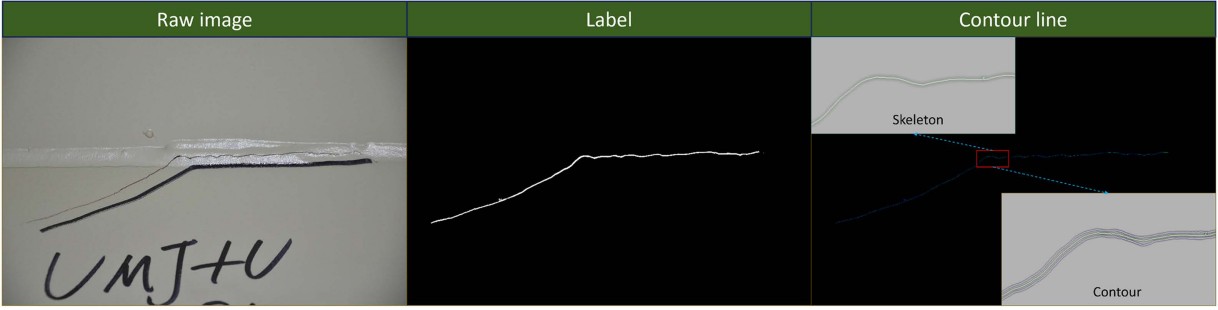

**Fig 8. The skeleton line of crack binary image.**

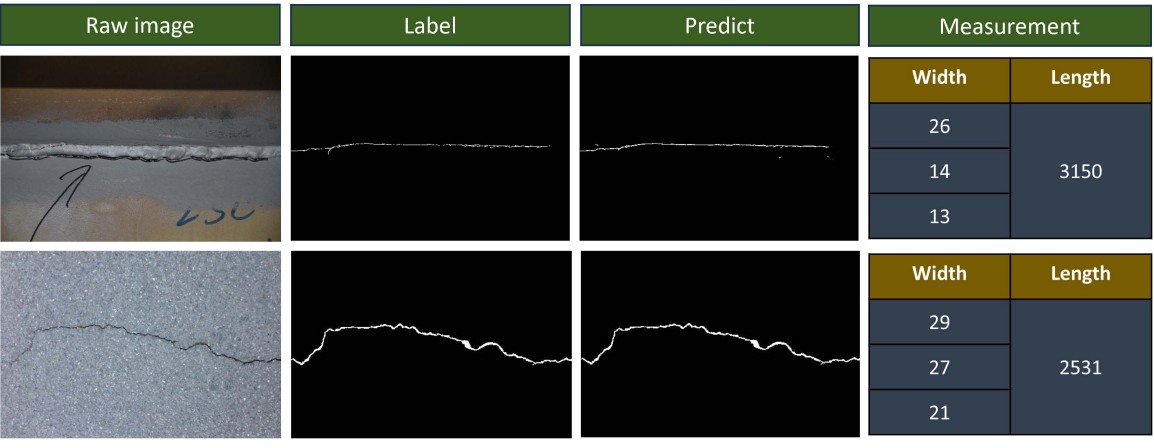

**Fig 9. Crack size information.**

quantifiable inputs for safety evaluation. To facilitate practical application, the detection process is designed to be compatible with commonly used high-resolution imaging technologies, such as digital single-lens reflex (DSLR) cameras, unmanned aerial vehicle (UAV)-mounted cameras, or mobile device-based imaging systems, depending on the inspection scenario. These images are then processed through the proposed computational pipeline to enable automated damage quantification. However, structural safety assessment does not rely solely on individual damage indicators; rather, it requires a comprehensive consideration of multiple damage features and their impact on overall structural stability. To address this, this study further developed a machine learning-based safety assessment method, which integrates multiple damage features and employs a RF algorithm to classify and predict structural safety conditions.

## Safety assessment data sources

To construct the structural safety assessment model and verify its effectiveness, this study collected 300 sets of structural damage data samples, covering various typical surface damage types, including cracks, concrete spalling, and exposed rebar.

The dataset in this study was primarily sourced from two parts. First, a portion of the data was obtained from real-world engineering structures, including bridges, tunnels, roads, and hydraulic infrastructure. These inspection images and field measurement data were collected using high-resolution imaging equipment and subsequently processed using the proposed damage detection method to extract features such as crack length, spalling area, and rebar exposure, ensuring data accuracy and engineering applicability. Second, to enhance dataset diversity and improve the model's generalization ability, this study incorporated publicly available datasets from relevant research papers, which were carefully standardized and processed to ensure consistency in damage characterization [75–77].

Each data sample in the dataset contains the following key attributes:

Crack features: Number, length, width, and direction.

Concrete spalling features: Spalling area.

Exposed rebar features: Presence or absence of exposed rebar

Structural material type: Concrete or steel structure.

Safety assessment labels: Expert evaluations or engineering code-based safety ratings. The safety assessment labels classify structural conditions into three levels: safe (S), minor damage (MD), severe damage (SD). The dataset consists of 190 samples for S, 70 for MD, and 40 for SD. These labels were determined at the element level by a panel of five structural experts following national standards, ensuring consistency through a consensus-voting mechanism to mitigate inter-rater variability. To ensure effective model training and reliable evaluation, the dataset was divided into training, testing, and validation sets in an 80:10:10 ratio.

To ensure effective model training and reliable evaluation, the dataset was divided into training, testing, and validation sets in an 80:10:10 ratio. The training set was used to optimize the mapping relationship between damage features and structural safety levels. The testing set was used to assess the generalization ability of the model, ensuring its applicability to unseen samples. The validation set was used for hyperparameter tuning, optimizing the training process, and preventing overfitting. Additionally, several preprocessing operations were applied to improve data quality and model performance, including normalization and missing value handling. For numerical features (e.g., crack length, width, and spalling area), the min-max normalization method is used to scale the values to the range of 0–1. This process can eliminate the differences between different physical units. For missing values, mean interpolation are used to ensure data integrity.

## Safety evaluation model and result analysis

**The random forest algorithm for structural safety assessment.** The RF algorithm was employed to construct the structural safety assessment model. The RF is an ensemble learning method based on decision trees, which improves classification accuracy and generalization ability through a voting mechanism among multiple decision trees. Compared to a single decision tree, RF demonstrates significant advantages in handling high-dimensional data, multi-feature inputs,

and nonlinear relationships. In structural surface damage analysis, damage feature data often exhibit nonlinear and complex coupling relationships, making it difficult to model using simple linear models. Random Forest can automatically select optimal features and construct multiple weak classifiers using the Bagging (bootstrap sampling) strategy, thereby reducing the risk of overfitting in individual models. Furthermore, RF is robust to outliers and can evaluate the importance of different features, which helps identify the dominant damage parameters in structural safety assessment.

This study uses seven types of damage features (number of cracks, total crack length, local maximum width, crack direction, concrete spalling area, exposed steel bar area, and material types) as input variables and safety rating labels (S, MD, and SD) as output categories to train a RF classification model. During the model training process, ten-fold cross validation was used to optimize hyperparameters, and 50 decision trees were finally selected. The maximum tree depth was optimized through grid search to ensure the stability of classification performance.

Based on the RF-based safety assessment model, the prediction results for 30 validation samples are illustrated in Fig 10. The confusion matrix indicates that the model performs well in evaluating the S category but exhibits certain mis-classification phenomena when distinguishing between MD and SD categories.

All samples classified as S were correctly identified, achieving a recall rate of 100%. This result suggests that the random forest model demonstrates exceptionally high accuracy in recognizing intact structures, effectively distinguishing undamaged structures from damaged ones. This experimental result verifies the powerful learning ability of RF for high-dimensional features.

For the MD category, a total of seven samples were evaluated, among which five were correctly classified, one was misclassified as S, and another was misclassified as SD. These misclassification cases may be attributed to the similarity of damage features, particularly when crack length is short or spalling area is small, as some minor damage characteristics may resemble those of intact structures, leading to misclassification as S. Additionally, certain minor damage samples may exhibit more pronounced damage features, causing the model to categorize them as SD.

For the SD category, three samples were evaluated, with two correctly classified and one misclassified as MD. This suggests that some severe damage features may not have been fully captured by the model, or that certain minor and severe damage cases share overlapping feature distributions, making precise differentiation challenging. For instance,

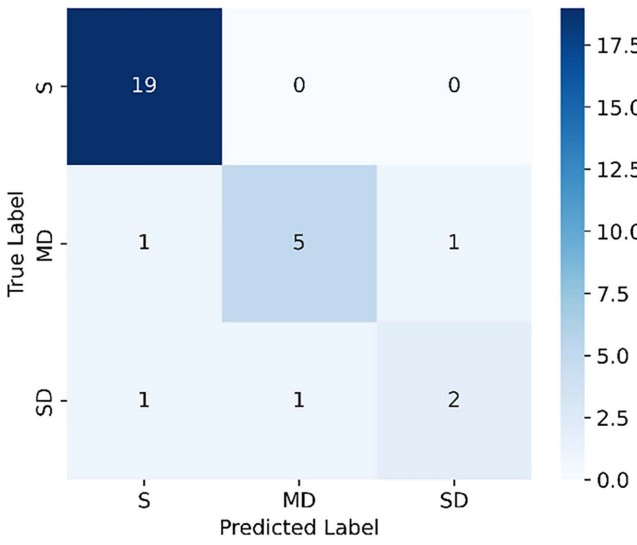

**Fig 10. Confusion matrix of safety evaluation.**

if the damage is spatially dispersed while the maximum local crack width does not exceed a critical threshold, the model may be inclined to classify the sample as MD rather than SD.

## Comparative experiment

To further verify the comprehensive performance of RF, we selected four mainstream machine learning algorithms for comparative analysis, namely gradient boosting (GB), support vector machine (SVM), multi-layer perceptron (MLP) and decision tree (DT). The optimal parameter configuration of each model is determined through pre-training, as shown in Table 3.

These models were trained on the same dataset and evaluated using the same performance indicators. To evaluate the classification performance of different models, accuracy, F1-score, and area under the curve (AUC) were selected as the primary evaluation metrics. The experimental results are presented in the Table 4.

Experimental results indicate significant differences in classification performance across different models. Among them, the RF model achieved the best performance on all evaluation metrics, with an accuracy of 87.0%, an F1-score of 0.76, and an AUC value of 0.83. These results suggest that the RF model can stably predict structural safety conditions and exhibits strong generalization capability. Ten-fold cross-validation further confirmed the robustness of the RF model, yielding a 95% confidence interval of [85.2%, 88.8%] for accuracy, which statistically significantly outperforms the comparison baselines.

As shown in Fig 11, confusion matrices of the GB, SVM, MLP, and DT models provide additional insights into their performance. The GB model (Fig 11a) demonstrated reasonable classification ability with moderate misclassification between adjacent safety categories, which aligns with its performance metrics (accuracy: 76.7%, F1-score: 0.59, AUC: 0.72). The SVM model (Fig 11b), in contrast, showed higher confusion, particularly in distinguishing SD and MD classes, resulting in the lowest overall performance among the models. The MLP model (Fig 11c) achieved slightly better results than SVM but still exhibited notable misclassification, particularly in the SD category, reflecting its reliance on larger datasets for effective learning. The DT model (Fig 11d) was found to be prone to overfitting, with a confusion matrix revealing inconsistent classification results and weaker generalization (accuracy: 73.3%, F1-score: 0.41, AUC: 0.62).

Overall, both the quantitative metrics and visual analysis of the confusion matrices confirm that the RF model is the most effective and robust approach for structural safety classification in this study.

**Table 3. Hyperparameter for each model.**

| Model | Learning Rate | Number of Trees/Layers | Max Depth | Min Samples | Model | Learning Rate | Number of Trees/Layers |
|-------|--------------|------------------------|-----------|-------------|-------|--------------|------------------------|
| RF | N/A | 50 | 8 | 2 | N/A | N/A | Ten-fold |
| GB | 0.05 | 30 | 6 | 20 | N/A | N/A | Ten -fold |
| SVM | N/A | N/A | N/A | N/A | N/A | N/A | SMO |
| MLP | 0.001 | (64, 32, 16) | N/A | N/A | 32 | 200 | Adam |
| DT | N/A | N/A | 10 | 2 | N/A | N/A | Ten -fold |

**Table 4. Comparative experiment.**

| Model | Accuracy (%) | F1-score | AUC |
|-------|-------------|----------|-----|
| RF | 87.0 | 0.76 | 0.83 |
| GB | 76.7 | 0.59 | 0.72 |
| SVM | 66.7 | 0.45 | 0.64 |
| MLP | 72.4 | 0.54 | 0.70 |
| DT | 73.3 | 0.41 | 0.62 |

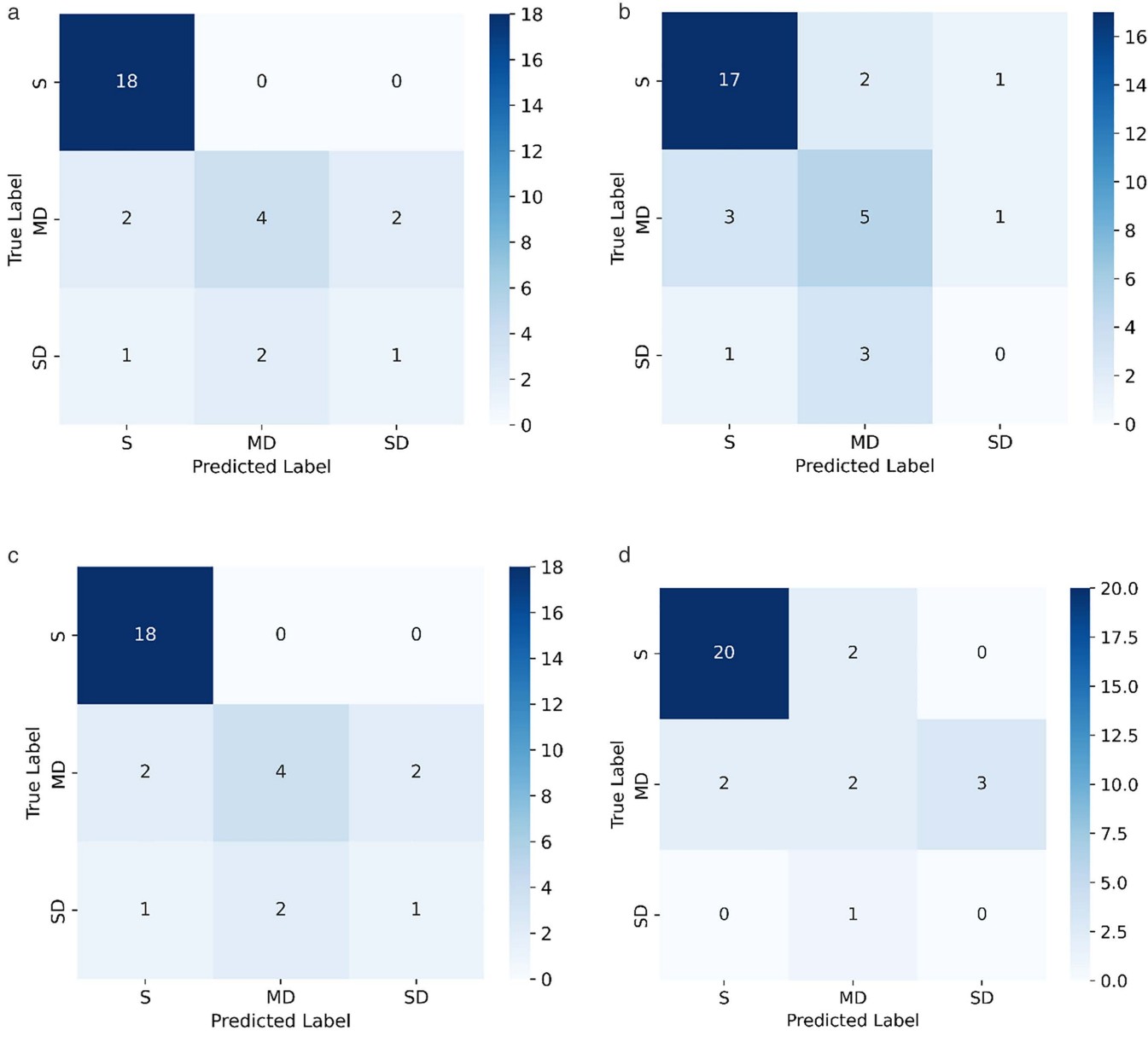

**Fig 11. Confusion matrix of several comparison models.** (a) GB. (b) SVM. (c) MLP. (d) DT.

## Conclusion

This study proposes a structural damage detection and safety assessment method that integrates machine vision and machine learning to enhance the accuracy of structural surface damage identification and the reliability of safety evaluation. Addressing the limitations of traditional detection methods, such as low efficiency and strong subjectivity, this study constructs a multi-scale damage detection framework that combines the ResNet-50 classification model and the SegFormer segmentation model to achieve efficient identification and quantitative analysis of surface damage, including cracks, concrete spalling, and exposed rebar. Furthermore, seven key damage indicators—material type, number of cracks, total crack length, maximum local crack width, spalling area, crack orientation, and rebar exposure—are extracted

to develop a data-driven safety assessment method based on the RF model. Comparative experiments with GB, SVM, MLP, and DT demonstrate that the RF model outperforms the other methods in terms of accuracy (87.0%), F1-score (0.76), and AUC (0.83), validating its effectiveness in handling complex damage data and improving the accuracy and generalization capability of safety assessments.

Despite the promising results, there is still room for improvement. It is important to note that surface-image features may not fully capture internal structural degradation; therefore, the proposed framework is intended as a rapid preliminary screening tool for large-scale infrastructure. Future work will focus on expanding and diversifying the dataset, particularly by incorporating damage data from extreme environments to enhance model robustness. Additionally, integrating multi-source sensor data (e.g., infrared, ultrasound, and laser scanning) for multimodal fusion analysis will further improve the comprehensiveness and precision of damage detection and safety assessment.

## Author contributions

**Conceptualization:** Shengmin Wang.

**Data curation:** Shengmin Wang.

**Formal analysis:** Shengmin Wang.

**Funding acquisition:** Di Le.

**Investigation:** Shengmin Wang.

**Methodology:** Shengmin Wang.

**Project administration:** Di Le.

**Resources:** Di Le.

**Supervision:** Di Le.

**Validation:** Shengmin Wang, Moxiao Li.

**Visualization:** Shengmin Wang.

**Writing – original draft:** Shengmin Wang.

**Writing – review & editing:** Moxiao Li, Di Le.

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
