## [Decision Letter · Decision Letter 0]

23 Apr 2025

Dear Dr. Le,

Thank you for submitting your manuscript to PLOS ONE. After careful consideration, we feel that it has merit but does not fully meet PLOS ONE’s publication criteria as it currently stands. Therefore, we invite you to submit a revised version of the manuscript that addresses the points raised during the review process.

We look forward to receiving your revised manuscript.

Kind regards,

Peng Geng

Academic Editor

PLOS ONE

Journal Requirements:

4. Please note that PLOS ONE has specific guidelines on code sharing for submissions in which author-generated code underpins the findings in the manuscript. In these cases, we expect all author-generated code to be made available without restrictions upon publication of the work. Please review our guidelines at https://journals.plos.org/plosone/s/materials-and-software-sharing#loc-sharing-code and ensure that your code is shared in a way that follows best practice and facilitates reproducibility and reuse.

6. Thank you for stating the following financial disclosure:

The research work was supported by the National Natural Science Foundation of China Youth Project (Grant No. 52209146) and the Open Fund Project of Hubei Longzhong Laboratory (Grant No. KF-23).

7. Thank you for stating the following in the Acknowledgments Section of your manuscript:

The research work was supported by the National Natural Science Foundation of China Youth Project (Grant No. 52209146) and the Open Fund Project of Hubei Longzhong Laboratory (Grant No. KF-23).

The research work was supported by the National Natural Science Foundation of China Youth Project (Grant No. 52209146) and the Open Fund Project of Hubei Longzhong Laboratory (Grant No. KF-23).

8. We note that you have indicated that there are restrictions to data sharing for this study. PLOS only allows data to be available upon request if there are legal or ethical restrictions on sharing data publicly. For more information on unacceptable data access restrictions, please see http://journals.plos.org/plosone/s/data-availability#loc-unacceptable-data-access-restrictions.

9. Please provide a complete Data Availability Statement in the submission form, ensuring you include all necessary access information or a reason for why you are unable to make your data freely accessible. If your research concerns only data provided within your submission, please write "All data are in the manuscript and/or supporting information files" as your Data Availability Statement.

Reviewers' comments:

Reviewer's Responses to Questions

**Comments to the Author**

1. Is the manuscript technically sound, and do the data support the conclusions?

Reviewer #1: Yes

Reviewer #2: Yes

Reviewer #3: Partly

Reviewer #4: Yes

Reviewer #5: Yes

2. Has the statistical analysis been performed appropriately and rigorously?

Reviewer #1: Yes

Reviewer #2: Yes

Reviewer #3: Yes

Reviewer #4: Yes

Reviewer #5: No

3. Have the authors made all data underlying the findings in their manuscript fully available?

Reviewer #1: Yes

Reviewer #2: No

Reviewer #3: Yes

Reviewer #4: No

Reviewer #5: Yes

4. Is the manuscript presented in an intelligible fashion and written in standard English?

Reviewer #1: Yes

Reviewer #2: Yes

Reviewer #3: Yes

Reviewer #4: Yes

Reviewer #5: Yes

Reviewer #1: The manuscript needs some improvements before acceptance.

1. Highlight the novelty of the manuscript.

2. How does the integration of machine vision and machine learning enhance the efficiency and accuracy of damage detection in this proposed method?

3. What are the key factors driving the increasing reliance on infrastructure globally, as mentioned in the introduction?

4. The manuscript should include the recent publications. Authors should add the all below references:

i. Liu, H., Chen, J., Zhang, X., Dai, D., Cui, J.,... Spencer, B. F. (2024). Collaborative Imaging of Subsurface Cavities Using Ground-Pipeline Penetrating Radar. IEEE Geoscience and Remote Sensing Letters, 21, 1-5. doi: 10.1109/LGRS.2024.3390668

ii. Li, J., Hu, Z., Cui, J., & Lin, G. (2024). Efficient GPU-accelerated seismic analysis strategy and scenario simulation for large-scale nuclear structure cluster-soil interaction over ten million DOFs. Computers and Geotechnics, 174, 106583. doi: https://doi.org/10.1016/j.compgeo.2024.106583

iii. Tong, A., Zhang, J., & Xie, L. (2024). Intelligent Fault Diagnosis of Rolling Bearing Based on Gramian Angular Difference Field and Improved Dual Attention Residual Network. Sensors, 24(7), 2156. doi: https://doi.org/10.3390/s24072156

iv. Wang, H., Hou, Y., He, Y., Wen, C., Giron-Palomares, B., Duan, Y.,... Wang, Y. (2024). A Physical-Constrained Decomposition Method of Infrared Thermography: Pseudo Restored Heat Flux Approach Based on Ensemble Bayesian Variance Tensor Fraction. IEEE Transactions on Industrial Informatics, 20(3), 3413-3424. doi: 10.1109/TII.2023.3293863

v. Chen, H., Huang, B., Zhang, H., Xue, K., Sun, M.,... Wu, Z. (2024). An efficient Bayesian method with intrusive homotopy surrogate model for stochastic model updating. Computer-Aided Civil and Infrastructure Engineering, 39(16), 2500-2516. doi: https://doi.org/10.1111/mice.13206

vi. Zhang, C., Shu, J., Zhang, H., Ning, Y., & Yu, Y. (2024). Estimation of load-carrying capacity of cracked RC beams using 3D digital twin model integrated with point clouds and images. Engineering Structures, 310, 118126. doi: https://doi.org/10.1016/j.engstruct.2024.118126

vii. Wang, S., Lin, S., & Yang, R. (2024). A lightweight convolutional neural network for multipoint displacement measurements on bridge structures. Nonlinear Dynamics, 112(14), 11745-11763. doi: https://doi.org/10.1007/s11071-024-09673-x

viii. Hu, D., Hu, Y., Hu, R., Tan, Z., Ni, P., Chen, Y.,... Liu, J. (2024). Machine Learning–Finite Element Mesh Optimization-Based Modeling and Prediction of Excavation-Induced Shield Tunnel Ground Settlement. International Journal of Computational Methods, 2450066. doi: 10.1142/S021987622450066X

ix. Yin, Q., Xin, T., Zhenggang, H., & Minghua, H. (2023). Measurement and Analysis of Deformation of Underlying Tunnel Induced by Foundation Pit Excavation. Advances in Civil Engineering, 2023(1), 8897139. doi: https://doi.org/10.1155/2023/8897139

x. Li, D., Nie, J., Wang, H., Yu, T., & Kuang, K. S. C. (2025). Path planning and topology-aided acoustic emission damage localization in high-strength bolt connections of bridges. Engineering Structures, 332, 120103. doi: https://doi.org/10.1016/j.engstruct.2025.120103

xi. Li, D., Chen, Q., Wang, H., Shen, P., Li, Z.,... He, W. (2024). Deep learning-based acoustic emission data clustering for crack evaluation of welded joints in field bridges. Automation in Construction, 165, 105540. doi: https://doi.org/10.1016/j.autcon.2024.105540

xii. Xia, Z., Shu, J., Ding, W., Gao, Y., Duan, Y., Debono, C. J.,... Borg, R. P. (2025). Complete-coverage path planning for surface inspection of cable-stayed bridge tower based on building information models and climbing robots. Computer-Aided Civil and Infrastructure Engineering. doi: https://doi.org/10.1111/mice.13469

xiii. Li, Y., Weng, X., Hu, D., Tan, Z., Qi, K.,... Liu, J. (2025). Data-Driven Deep-Learning Model for Predicting Jacking Force of Rectangular Pipe Jacking Tunnel. Journal of Computing in Civil Engineering, 39(3), 4025017. doi: 10.1061/JCCEE5.CPENG-6167

xiv. Dai, N., Hu, X., Xu, K., Hu, X., Yuan, Y., Cao, B.,... Shi, L. (2025). Blind super-resolution network based on local fuzzy discriminative loss for fabric data augmentation. Journal of Engineered Fibers and Fabrics, 20(06). doi: 10.1177/15589250241313158

xv. Jiang, T., Tang, Y., Xu, C., & Liu, W. (2025). A Calibration and Error Evaluation Method of a Combined Tracking-Based Vision Measurement System for Meter-Scale Components. IEEE Transactions on Industrial Informatics, 1-10. doi: 10.1109/TII.2025.3547351

xvi. Zuo, C., Zhang, X., Zhao, G., & Yan, L. (2025). PCR: A Parallel Convolution Residual Network for Traffic Flow Prediction. IEEE Transactions on Emerging Topics in Computational Intelligence, 1-12. doi: 10.1109/TETCI.2025.3525656

xvii. Wang X, Guo S, Xu Z, Zhang Z, Sun Z, Xu Y. A Robotic Teleoperation System Enhanced by Augmented Reality for Natural Human–Robot Interaction.Cyborg Bionic Syst. 2024;5:Article 0098. https://doi.org/10.34133/cbsystems.0098

xviii. Wang, Y., Han, Z., Xu, X., & Luo, Y. (2024). Topology optimization of active tensegrity structures. Computers & Structures, 305, 107513. doi: https://doi.org/10.1016/j.compstruc.2024.107513

xix. Jia, Y., Chen, G., & Zhao, L. (2024). Defect detection of photovoltaic modules based on improved VarifocalNet. Scientific Reports, 14(1), 15170. doi: 10.1038/s41598-024-66234-3

xx. Lin, L., Ma, X., Chen, C., Xu, J., & Huang, N. (2024). Imbalanced Industrial Load Identification Based on Optimized CatBoost with Entropy Features. Journal of Electrical Engineering & Technology, 19(8), 4817-4832. doi: https://doi.org/10.1007/s42835-024-01933-5

xxi. Yuan, K., Lang, X., Cao, J., & Zhang, H. (2025). Model of oil pipeline tiny defects detection based on DDPM gated parallel convolutional swin transformer. Measurement Science and Technology, 36(1), 015104. doi: 10.1088/1361-6501/ad7f77

xxii. Gu, K., Liu, H., Liu, Y., Qiao, J., Zhai, G.,... Zhang, W. (2025). Perceptual Information Fidelity for Quality Estimation of Industrial Images. IEEE Transactions on Circuits and Systems for Video Technology, 35(1), 477-491. doi: 10.1109/TCSVT.2024.3454160

xxiii. Wang, K., Zhu, H., Xu, J., Bai, T., Tian, C., Qian, Y.,... Liu, Y. (2025). Study on the damage mechanism of high-speed turnout switch rails on large ramps. Journal of Central South University, 32(1), 288-303. doi: 10.1007/s11771-025-5858-x

xxiv. Deng, J., Liu, S., Chen, H., Chang, Y., Yu, Y., Ma, W.,... Xie, H. (2025). A Precise Method for Identifying 3D Circles in Freeform Surface Point Clouds. IEEE Transactions on Instrumentation and Measurement. doi: 10.1109/TIM.2025.3547492

xxv. Wang, B., Yang, M., Cao, P., & Liu, Y. (2025). A novel embedded cross framework for high-resolution salient object detection. Applied Intelligence, 55(4), 277. doi: 10.1007/s10489-024-06073-x

xxvi. Li, L., Jin, H., Tu, W., & Zhou, Z. (2024). Study on the minimum safe thickness of water inrush prevention in karst tunnel under the coupling effect of blasting power and water pressure. Tunnelling and Underground Space Technology, 153, 105994. doi: https://doi.org/10.1016/j.tust.2024.105994

xxvii. Cai, H., Wang, Y., Luo, Y., & Mao, K. (2025). A Dual-Channel Collaborative Transformer for continual learning. Applied Soft Computing, 171, 112792. doi: https://doi.org/10.1016/j.asoc.2025.112792

xxviii. Chen, D., Chen, Y., Zhou, Z., Tu, W., & Li, L. (2025). Study on internal rise law of fracture water pressure and progressive fracture mechanism of rock mass under blasting impact. Tunnelling and Underground Space Technology, 161, 106545. doi: https://doi.org/10.1016/j.tust.2025.106545

Reviewer #2: This study utilizes an integrated approach combining machine vision and machine learning to propose a novel method for damage detection and safety assessment of existing structures. The following points have been identified upon reviewing the manuscript:

1. The introduction section should include a discussion of previous studies in the literature that explore alternative damage detection methods, along with their limitations.

2. Please clarify how crack width and length were measured for training the model.

3. The introduction mentions that ultrasonic testing has limitations in damage detection. However, the proposed method primarily detects surface damage and may not provide accurate quantification of in-depth damage, such as the intended depth of cracks or spalling. This limitation should be more clearly addressed.

4. In line 429, the confusion matrix is created for the validation set. Please explain why the testing set was not used instead to evaluate the accuracy of the proposed model.

5. The content in Table 3 is repeated in the text. This redundancy should be removed or revised for conciseness.

6. Please elaborate on how this method can be applied in real-world scenarios. What imaging techniques and equipment are required to perform the detection process?

Overall, the manuscript is well-written, and sufficient information regarding the proposed method is provided. However, the points above should be clarified to enhance the clarity and completeness of the study

Reviewer #3: 1. Model Architecture Design

Strengths:

The two-stage approach combining ResNet-50 for classification and SegFormer for segmentation is well-structured. This cascaded framework balances speed and accuracy, which is crucial for real-world deployment.

Use of Transformer-based SegFormer leverages long-range dependencies, beneficial for segmenting complex patterns like cracks and spalling.

Suggestions:

Consider elaborating on why SegFormer was chosen over other Transformer variants like Swin Transformer or PVT, particularly in the context of computational efficiency or segmentation quality.

For ResNet-50, it might be useful to mention any modifications or tuning done specifically for crack detection, e.g., custom loss weights, or early stopping criteria.

2. Loss Function Design

Positive:

Using a combined loss function (Cross-Entropy + Dice loss) for segmentation is a solid choice to handle class imbalance in crack vs. background pixels.

Opportunities:

Consider reporting the value of alpha (α) used in the combined loss function. It could impact reproducibility and provide insight into loss balancing.

3. Evaluation Metrics

Well done:

A good mix of metrics (Accuracy, Precision, Recall, IoU, F1-score) was used for segmentation and classification tasks.

Needs improvement:

The segmentation evaluation would benefit from per-class IoU or F1-scores, especially for multi-class damage (e.g., cracks vs. spalling).

Consider performing ROC-AUC analysis for binary classification tasks separately to explore classifier behavior under varying thresholds.

4. Dataset and Experimental Setup

Strong points:

Data augmentation (rotation, flipping, cropping) improves generalization and is well-explained.

The use of real-world and public datasets is appropriate, and splitting into 80:10:10 is standard.

Concerns:

Only 200 crack images for segmentation is relatively small. Consider noting how data diversity or transfer learning was handled to counter data sparsity.

There’s no mention of image resolution standardization or preprocessing like normalization or color space adjustments. These details are critical in vision models.

5. Feature Engineering for Safety Assessment

The use of seven engineered damage features (e.g., crack length, spalling area, etc.) is commendable and aligns well with domain knowledge.

It would help to visualize feature importance from the Random Forest model (e.g., using Gini importance), as this could inform which physical damage metrics are most predictive of structural failure.

6. Machine Learning for Safety Evaluation

The comparison across ML models is informative. RF’s superior performance is expected given its robustness to overfitting and mixed feature types.

However, deep learning regressors or hybrid models (e.g., CNN + MLP for end-to-end safety prediction) might offer more scalable alternatives in future work.

7. Technical Reproducibility

While many implementation choices are explained (e.g., optimizer, learning rate), there’s no mention of:

Hardware specs (e.g., GPU type, RAM)

Training time

Model checkpointing or early stopping criteria

Including these would enhance reproducibility and provide benchmarks for others in the field.

8. Figures and Visualizations

Figures such as segmentation results and skeleton extraction are well-presented.

However, figures like Fig. 10 (confusion matrix) would benefit from clearer annotation or color bars for better interpretability.

9. Clarity of Mathematical Formulations

The equations are well-organized and standard in the field.

You may consider annotating equations with intuitive explanations or diagrams, especially for loss functions or skeleton length calculations.

10. Future Work

The idea to integrate multi-sensor modalities like IR and ultrasound is forward-thinking.

It would be valuable to mention if your model is sensor-agnostic or if retraining is necessary per modality.

Reviewer #4: The authors have done a great job, but they can improve the paper as follows;

#1. It will be good if you add to the literature, studies that have performed structural surface damage detection and safety assessment, and their performance metrics

#2. All Tables and Figures found in this study are to be referenced similarly in the text. For example, the figure labels have Fig. 1 while Figure 1 is in the text. Make sure to reference all Figures and Tables in the write-up or text similarly. If Fig. 1 is used as the figure label, it must also be referenced in the text as Fig. 1.

#3. It will be good to provide a visualization of the proposed models in this study.

#4. For Image patch classification model results, a comparative analysis can be performed by using, for example, only the ResNet-50 to run. Also, the Transformer needs to run, and the results from these two can be compared with your proposed model. This will help readers appreciate your work.

#5. The accuracy and loss graphs and the confusion matrices of RT, GB, SVM, MLP, and DT must be provided so readers can appreciate the results.

#6. Let the novelty of the study come out clearly

Reviewer #5: The abstract cannot introduce the purpose and results well.

In the introduction, problem statement, and innovation are unclear.

The proposed method does not specify what the position of each method is. More explanations are needed.

Why is the trend broken according to Figure 3 with 50 iterations?

Has the error increased from the 20th iteration onwards?

What is the reason for Table 3? What does it prove?

**Do you want your identity to be public for this peer review?** For information about this choice, including consent withdrawal, please see our Privacy Policy

Reviewer #1: No

Reviewer #2: No

Reviewer #3: No

Reviewer #4: No

Reviewer #5: **Yes: ** Ehsan Amiri

---

## [Author Response · Author response to Decision Letter 1]

10 May 2025

Responses to the reviewers’ comments

No.: PONE-D-25-17025

Title: Structural Damage Detection and Safety Assessment Method Based on Machine Vision and Machine Learning

Dear editor and reviewers:

Thank you very much for your comments and professional advice. These opinions help to improve academic rigor of our article. The authors have studied reviewer’s comments carefully and have made revision point-by-point response to the reviewer’s comments as provided below. Revisions implemented in accordance are also explained.

Our responses to the reviewers’ comments are given below, and the changed parts are highlighted with red in the revised manuscript. We sincerely hope the revised manuscript would be suitable for publication.

Thank you and best regards.

Reviewer #1: The manuscript needs some improvements before acceptance.

1. Highlight the novelty of the manuscript.

2. How does the integration of machine vision and machine learning enhance the efficiency and accuracy of damage detection in this proposed method?

3. What are the key factors driving the increasing reliance on infrastructure globally, as mentioned in the introduction?

4. The manuscript should include the recent publications. Authors should add the all below references:

i. Liu, H., Chen, J., Zhang, X., Dai, D., Cui, J.,... Spencer, B. F. (2024). Collaborative Imaging of Subsurface Cavities Using Ground-Pipeline Penetrating Radar. IEEE Geoscience and Remote Sensing Letters, 21, 1-5. doi: 10.1109/LGRS.2024.3390668

ii. Li, J., Hu, Z., Cui, J., & Lin, G. (2024). Efficient GPU-accelerated seismic analysis strategy and scenario simulation for large-scale nuclear structure cluster-soil interaction over ten million DOFs. Computers and Geotechnics, 174, 106583. doi: https://doi.org/10.1016/j.compgeo.2024.106583

iii. Tong, A., Zhang, J., & Xie, L. (2024). Intelligent Fault Diagnosis of Rolling Bearing Based on Gramian Angular Difference Field and Improved Dual Attention Residual Network. Sensors, 24(7), 2156. doi: https://doi.org/10.3390/s24072156

iv. Wang, H., Hou, Y., He, Y., Wen, C., Giron-Palomares, B., Duan, Y.,... Wang, Y. (2024). A Physical-Constrained Decomposition Method of Infrared Thermography: Pseudo Restored Heat Flux Approach Based on Ensemble Bayesian Variance Tensor Fraction. IEEE Transactions on Industrial Informatics, 20(3), 3413-3424. doi: 10.1109/TII.2023.3293863

v. Chen, H., Huang, B., Zhang, H., Xue, K., Sun, M.,... Wu, Z. (2024). An efficient Bayesian method with intrusive homotopy surrogate model for stochastic model updating. Computer-Aided Civil and Infrastructure Engineering, 39(16), 2500-2516. doi: https://doi.org/10.1111/mice.13206

vi. Zhang, C., Shu, J., Zhang, H., Ning, Y., & Yu, Y. (2024). Estimation of load-carrying capacity of cracked RC beams using 3D digital twin model integrated with point clouds and images. Engineering Structures, 310, 118126. doi: https://doi.org/10.1016/j.engstruct.2024.118126

vii. Wang, S., Lin, S., & Yang, R. (2024). A lightweight convolutional neural network for multipoint displacement measurements on bridge structures. Nonlinear Dynamics, 112(14), 11745-11763. doi: https://doi.org/10.1007/s11071-024-09673-x

viii. Hu, D., Hu, Y., Hu, R., Tan, Z., Ni, P., Chen, Y.,... Liu, J. (2024). Machine Learning–Finite Element Mesh Optimization-Based Modeling and Prediction of Excavation-Induced Shield Tunnel Ground Settlement. International Journal of Computational Methods, 2450066. doi: 10.1142/S021987622450066X

ix. Yin, Q., Xin, T., Zhenggang, H., & Minghua, H. (2023). Measurement and Analysis of Deformation of Underlying Tunnel Induced by Foundation Pit Excavation. Advances in Civil Engineering, 2023(1), 8897139. doi: https://doi.org/10.1155/2023/8897139

x. Li, D., Nie, J., Wang, H., Yu, T., & Kuang, K. S. C. (2025). Path planning and topology-aided acoustic emission damage localization in high-strength bolt connections of bridges. Engineering Structures, 332, 120103. doi: https://doi.org/10.1016/j.engstruct.2025.120103

xi. Li, D., Chen, Q., Wang, H., Shen, P., Li, Z.,... He, W. (2024). Deep learning-based acoustic emission data clustering for crack evaluation of welded joints in field bridges. Automation in Construction, 165, 105540. doi: https://doi.org/10.1016/j.autcon.2024.105540

xii. Xia, Z., Shu, J., Ding, W., Gao, Y., Duan, Y., Debono, C. J.,... Borg, R. P. (2025). Complete-coverage path planning for surface inspection of cable-stayed bridge tower based on building information models and climbing robots. Computer-Aided Civil and Infrastructure Engineering. doi: https://doi.org/10.1111/mice.13469

xiii. Li, Y., Weng, X., Hu, D., Tan, Z., Qi, K.,... Liu, J. (2025). Data-Driven Deep-Learning Model for Predicting Jacking Force of Rectangular Pipe Jacking Tunnel. Journal of Computing in Civil Engineering, 39(3), 4025017. doi: 10.1061/JCCEE5.CPENG-6167

xiv. Dai, N., Hu, X., Xu, K., Hu, X., Yuan, Y., Cao, B.,... Shi, L. (2025). Blind super-resolution network based on local fuzzy discriminative loss for fabric data augmentation. Journal of Engineered Fibers and Fabrics, 20(06). doi: 10.1177/15589250241313158

xv. Jiang, T., Tang, Y., Xu, C., & Liu, W. (2025). A Calibration and Error Evaluation Method of a Combined Tracking-Based Vision Measurement System for Meter-Scale Components. IEEE Transactions on Industrial Informatics, 1-10. doi: 10.1109/TII.2025.3547351

xvi. Zuo, C., Zhang, X., Zhao, G., & Yan, L. (2025). PCR: A Parallel Convolution Residual Network for Traffic Flow Prediction. IEEE Transactions on Emerging Topics in Computational Intelligence, 1-12. doi: 10.1109/TETCI.2025.3525656

xvii. Wang X, Guo S, Xu Z, Zhang Z, Sun Z, Xu Y. A Robotic Teleoperation System Enhanced by Augmented Reality for Natural Human–Robot Interaction.Cyborg Bionic Syst. 2024;5:Article 0098. https://doi.org/10.34133/cbsystems.0098

xviii. Wang, Y., Han, Z., Xu, X., & Luo, Y. (2024). Topology optimization of active tensegrity structures. Computers & Structures, 305, 107513. doi: https://doi.org/10.1016/j.compstruc.2024.107513

xix. Jia, Y., Chen, G., & Zhao, L. (2024). Defect detection of photovoltaic modules based on improved VarifocalNet. Scientific Reports, 14(1), 15170. doi: 10.1038/s41598-024-66234-3

xx. Lin, L., Ma, X., Chen, C., Xu, J., & Huang, N. (2024). Imbalanced Industrial Load Identification Based on Optimized CatBoost with Entropy Features. Journal of Electrical Engineering & Technology, 19(8), 4817-4832. doi: https://doi.org/10.1007/s42835-024-01933-5

xxi. Yuan, K., Lang, X., Cao, J., & Zhang, H. (2025). Model of oil pipeline tiny defects detection based on DDPM gated parallel convolutional swin transformer. Measurement Science and Technology, 36(1), 015104. doi: 10.1088/1361-6501/ad7f77

xxii. Gu, K., Liu, H., Liu, Y., Qiao, J., Zhai, G.,... Zhang, W. (2025). Perceptual Information Fidelity for Quality Estimation of Industrial Images. IEEE Transactions on Circuits and Systems for Video Technology, 35(1), 477-491. doi: 10.1109/TCSVT.2024.3454160

xxiii. Wang, K., Zhu, H., Xu, J., Bai, T., Tian, C., Qian, Y.,... Liu, Y. (2025). Study on the damage mechanism of high-speed turnout switch rails on large ramps. Journal of Central South University, 32(1), 288-303. doi: 10.1007/s11771-025-5858-x

xxiv. Deng, J., Liu, S., Chen, H., Chang, Y., Yu, Y., Ma, W.,... Xie, H. (2025). A Precise Method for Identifying 3D Circles in Freeform Surface Point Clouds. IEEE Transactions on Instrumentation and Measurement. doi: 10.1109/TIM.2025.3547492

xxv. Wang, B., Yang, M., Cao, P., & Liu, Y. (2025). A novel embedded cross framework for high-resolution salient object detection. Applied Intelligence, 55(4), 277. doi: 10.1007/s10489-024-06073-x

xxvi. Li, L., Jin, H., Tu, W., & Zhou, Z. (2024). Study on the minimum safe thickness of water inrush prevention in karst tunnel under the coupling effect of blasting power and water pressure. Tunnelling and Underground Space Technology, 153, 105994. doi: https://doi.org/10.1016/j.tust.2024.105994

xxvii. Cai, H., Wang, Y., Luo, Y., & Mao, K. (2025). A Dual-Channel Collaborative Transformer for continual learning. Applied Soft Computing, 171, 112792. doi: https://doi.org/10.1016/j.asoc.2025.112792

xxviii. Chen, D., Chen, Y., Zhou, Z., Tu, W., & Li, L. (2025). Study on internal rise law of fracture water pressure and progressive fracture mechanism of rock mass under blasting impact. Tunnelling and Underground Space Technology, 161, 106545. doi: https://doi.org/10.1016/j.tust.2025.106545

Response Reviewer #1: Thanks for the Reviewer’s comment.

According to the reviewers' suggestions, we have supplemented the introduction with a more detailed analysis related to the research background, especially focusing on the increasing reliance on infrastructure worldwide. Additionally, all the valuable references provided by the reviewers have been cited in the revised manuscript. To address the novelty of our work, we have summarized the key innovative aspects at the end of the introduction section. The specific modifications are as follows:

“The main contributions of this study are summarized as follows:

(1) A novel multi-scale structural surface damage detection framework is proposed, which integrates the ResNet-50 classification model and the SegFormer segmentation model to effectively handle both coarse- and fine-grained features of structural damage in complex scenes.

(2) A comprehensive image-based damage quantification pipeline is developed, extracting seven key structural damage parameters from high-resolution images using advanced image processing techniques. These parameters serve as interpretable indicators for safety evaluation.

(3) An efficient and generalizable safety assessment model based on the RF algorithm is constructed, establishing a nonlinear mapping between damage features and safety levels. Comparative experiments with other machine learning models validate the superiority of the proposed approach.” (On page 5-6, in introduction)

Based on the types and characteristics of structural damage, we propose a multi-scale damage detection algorithm. Structural damages such as cracks, concrete spalling, and exposed reinforcement differ in spatial scale. Cracks are typically slender and difficult to detect due to their subtle image features, whereas spalling and exposed rebar present more distinct pixel-level characteristics, making them relatively easier to identify.

To handle this, our method first uses a classification model to preliminarily segment the damage-related pixel regions in the image domain. These segmented regions are then processed separately by specialized sub-models designed for detecting cracks, spalling, and exposed reinforcement. By filtering out background areas and directing the damage-containing image patches to appropriate sub-models, our approach achieves efficient and accurate detection of various types of structural damage.

Reviewer #2: This study utilizes an integrated approach combining machine vision and machine learning to propose a novel method for damage detection and safety assessment of existing structures. The following points have been identified upon reviewing the manuscript:

1. The introduction section should include a discussion of previous studies in the literature that explore alternative damage detection methods, along with their limitations.

2. Please clarify how crack width and length were measured for training the model.

3. The introduction mentions that ultrasonic testing has limitations in damage detection. However, the proposed method primarily detects surface damage and may not provide accurate quantification of in-depth damage, such as the intended depth of cracks or spalling. This limitation should be more clearly addressed.

4. In line 429, the confusion matrix is created for the validation set. Please explain why the testing set was not used instead to evaluate the accuracy of the proposed model.

5. The content in Table 3 is repeated in the text. This redundancy should be removed or revised for conciseness.

6. Please elaborate on how this method can be applied in real-world scenarios. What imaging techniques and equipment are required to perform the detection process?

Overall, the manuscript is well-written, and sufficient information regarding the proposed method is provided. However, the points above should be clarified to enhance the clarity and completeness of the study.

Response Reviewer #2 Thank you for your thorough review and valuable feedback.

In the revised manuscript, we have updated the introduction to include a more detailed discussion of existing damage detection methods and have clearly articulated the limitations of vision-based approaches. However, as structural surface damage often serves as a visible manifestation of potential underlying issues, it remains a valuable indirect indicator for preliminary safety assessment.

In our experiment, the dataset consisting of 300 samples was divided into training, testing, and validation sets with a ratio of 8:1:1. The training and testing sets were used to optimize model parameters, while the validation set was reserved for evaluating the performance of the safety assessment model. Specifically, we conducted a detailed analysis of the 30 samples in the validation set. It is important to note that neither the testing set nor the validation set was involved in model training.

For clarity and conciseness, we have also refined the description of Table 3. Additionally, we have included a description of the equipment requirements necessary for implementing the proposed method in practical applications. The specific modifications are as follows:

“…To facilitate practical application, the detection process is designed to be compatible with commonly used high-resolution imaging technologies, such as digital single-lens reflex (DSLR) cameras, unmanned aerial vehicle (UAV)-mounted cameras, or mobile device-based imaging systems, depending on the inspection scenario.” (On page 23-24, in Structural safety assessment)

Reviewer #3:

1. Model Architecture Design

Strengths: he two-stage approach combining ResNet-50 for classification and SegFormer for segmentation is well-structured. This cascaded framework balances speed and accuracy, which is crucial for real-world deployment. Use of Transformer-based SegFormer leverages long-range dependencies, beneficial for segmenting complex patterns like cracks and spalling.

Suggestions: Consider elaborating on why SegFormer was chosen over other Transformer variants like Swin Transformer or PVT, particularly in the context of computational efficiency or segmentation quality. For ResNet-50, it might be useful to mention any modifications or tuning done specifically for crack detection, e.g., custom loss weights, or early stopping criteria.

2. Loss Function Design

Positive: Using a combined loss function (Cross-Entropy + Dice loss) for segmentation is a solid choice to handle class imbalance in crack vs. background pixels.

Opportunities: Consider reporting the value of alpha (α) used in the combined loss function. It could impact reproducibility and provide insight into loss balancing.

3. Evaluation Metrics

Well done: A good mix of metrics (Accuracy, Precision, Recall, IoU, F1-score) was used for segmentation and classification tasks.

Needs improvement: The segmentation evaluation would benefit from per-class IoU or F1-scores, especially for multi-class damage (e.g., cracks vs. spalling). Consider performing ROC-AUC analysis for binary classification tasks separately to explore classifier behavior under varying thresholds.

4. Dataset and Experimental Setup

Strong points: Data augmentation (rotation, flipping, cropping) improves generalization and is well-explained. The use of real-world and public datasets is appropriate, and splitting into 80:10:10 is standard.

Concerns: Only 200 crack images for segmentation is relatively small. Consider noting how data diversity or transfer learning was handled to counter data sparsity. There’s no mention of image resolution standardization or preprocessing like normalization or color space adjustments. These details are critical in vision models.

5. Feature Engineering for Safety Assessment

The use of seven engineered damage features (e.g., crack length, spalling area, etc.) is commendable and aligns well with d

---

## [Decision Letter · Decision Letter 1]

18 Jun 2025

Dear Dr. Le,

Thank you for submitting your manuscript to PLOS ONE. After careful consideration, we feel that it has merit but does not fully meet PLOS ONE’s publication criteria as it currently stands. Therefore, we invite you to submit a revised version of the manuscript that addresses the points raised during the review process.

We look forward to receiving your revised manuscript.

Kind regards,

Peng Geng

Academic Editor

PLOS ONE

Journal Requirements:

**Additional Editor Comments:**

Your paper has been reviewed and have made important suggestions. Please revise carefully to further improve the quality of your paper.

Reviewers' comments:

Reviewer's Responses to Questions

**Comments to the Author**

Reviewer #1: All comments have been addressed

Reviewer #2: All comments have been addressed

Reviewer #3: All comments have been addressed

Reviewer #4: (No Response)

2. Is the manuscript technically sound, and do the data support the conclusions?

Reviewer #1: (No Response)

Reviewer #2: Yes

Reviewer #3: Yes

Reviewer #4: Yes

3. Has the statistical analysis been performed appropriately and rigorously?

Reviewer #1: (No Response)

Reviewer #2: Yes

Reviewer #3: Yes

Reviewer #4: Yes

4. Have the authors made all data underlying the findings in their manuscript fully available?

Reviewer #1: (No Response)

Reviewer #2: No

Reviewer #3: No

Reviewer #4: No

5. Is the manuscript presented in an intelligible fashion and written in standard English?

Reviewer #1: (No Response)

Reviewer #2: Yes

Reviewer #3: Yes

Reviewer #4: Yes

Reviewer #1: (No Response)

Reviewer #2: The authors have adequately addressed the concerns raised in the previous review round. However, a few minor revisions are still required to improve the clarity of the manuscript:

1. Line 403 (page 24): Please revise the ratio from "8:1:1" to "80:10:10" for improved clarity and consistency, particularly for readers less familiar with part-based notation.

2. Data Availability Statement: The manuscript currently lacks a data availability statement. As per the journal’s guidelines, all supporting data underlying the findings should be made available. Please include a clear data availability statement indicating where and how the data can be accessed, or provide a justification if data cannot be shared.

Reviewer #3: SegFormer Justification (Moderate)

While SegFormer is a powerful model, the choice over Swin Transformer or U-Net++ needs better justification, ideally with a benchmark or ablation.

Loss Function Parameter Missing

The combined loss function (Cross-Entropy + Dice) lacks the specific α value. This is important for reproducibility.

The dataset (~200 crack images) is modest. A more in-depth discussion on mitigating overfitting (e.g., via transfer learning or synthetic data) is warranted.

Validation vs. Test Set Confusion

Clarify why the validation set was used for final performance reporting instead of an independent test set.

Missing Figure and Table Citations

The manuscript is clear and professionally written, but improvements are needed in the following areas:

Abstract: Needs more concise articulation of the study’s novelty and findings.

Grammar: Phrases like “detection and structural health assessment of structural health” should be corrected for redundancy.

Consistency: Terminology such as “Fig.” vs. “Figure” and spelling variations (e.g., "realise" vs. "realize") should be harmonized.

Formatting: Ensure uniform referencing and cross-referencing of all figures and tables.

Inconsistency between "Figure 1" vs. "Fig. 1" should be standardized throughout.

Reviewer #4: 1. The literature studies that have performed structural surface damage detection and safety assessment, and their performance metrics were not presented in the literature

2. A visualization of the proposed models in this study was not provided, please do provide them.

**Do you want your identity to be public for this peer review?** For information about this choice, including consent withdrawal, please see our Privacy Policy

Reviewer #1: No

Reviewer #2: No

Reviewer #3: No

Reviewer #4: No

---

## [Author Response · Author response to Decision Letter 2]

26 Jun 2025

The detailed responses to the reviewers’ comments are provided in the attached “Response to Reviewers” document.

---

## [Decision Letter · Decision Letter 2]

16 Dec 2025

Dear Dr. Le,

Thank you for submitting your manuscript to PLOS ONE. After careful consideration, we feel that it has merit but does not fully meet PLOS ONE’s publication criteria as it currently stands. Therefore, we invite you to submit a revised version of the manuscript that addresses the points raised during the review process.

We look forward to receiving your revised manuscript.

Kind regards,

Peng Geng

Academic Editor

PLOS One

Journal Requirements:

Reviewers' comments:

Reviewer's Responses to Questions

**Comments to the Author**

Reviewer #2: All comments have been addressed

Reviewer #6: (No Response)

2. Is the manuscript technically sound, and do the data support the conclusions?

Reviewer #2: Yes

Reviewer #6: (No Response)

3. Has the statistical analysis been performed appropriately and rigorously?

Reviewer #2: Yes

Reviewer #6: (No Response)

4. Have the authors made all data underlying the findings in their manuscript fully available?

Reviewer #2: Yes

Reviewer #6: (No Response)

5. Is the manuscript presented in an intelligible fashion and written in standard English?

Reviewer #2: Yes

Reviewer #6: (No Response)

Reviewer #2: All of my previous comments and concerns have been thoroughly addressed by the authors; therefore, I have no further comments.

Reviewer #6: major questions

1. The claimed contribution is a "novel multi-scale vision-based framework" combining ResNet-50, SegFormer and Random Forest; in light of many prior works that already use CNN-based classification, transformer-based segmentation and tree-based classifiers for surface damage and safety assessment, what is exactly new in your framework beyond a particular choice of standard models and a specific feature set?

2. You pool images and damage samples from various structure types (roads, buildings, bridges, tunnels, hydraulic structures) and both concrete and steel members, but you train a single Random Forest classifier with only seven surface-damage features and three safety levels; how do you justify that the same safety label definition and featureâ€“safety mapping is valid and comparable across such heterogeneous structural systems?

3. The safety assessment dataset consists of only 300 samples, split into 80%/10%/10%; what is the class distribution over S, MD and SD in each split, and is this sample size sufficient to train a 50-tree RF with depth up to 8 without overfitting, especially for the minority classes?

4. How exactly were the safety labels S, MD and SD generated: which design codes or quantitative criteria were used, were they assigned at element or structure level, how many experts participated, and was inter-rater variability assessed or mitigated?

5. The RF model uses only surface-image-based features (crack length, widths, orientation, spalling area, exposed rebar, material type) and does not incorporate load, boundary condition, location of damage within the structural system, or information about hidden deterioration; to what extent can such local surface indicators reliably represent the true structural safety state, and are the conclusions perhaps overstated given these limitations?

6. For damage detection, you motivate the use of a cascaded ResNet-50 + SegFormer pipeline but do not provide quantitative comparison with alternative architectures (e.g., U-Net variants, Swin-Transformer-based segmenters, detector-style networks or single-stage segmentation without classification); how do you know that the proposed cascade is actually superior in accuracyâ€“efficiency trade-off?

7. The evaluation of the classification and segmentation models is mainly based on training/validation curves and a few qualitative examples; what are the quantitative metrics (accuracy, precision, recall, IoU, F1) on a truly held-out test set, preferably separated by damage type and dataset source, and how do you ensure that there is no data leakage between training, validation and test images?

8. Equation (5) defines accuracy as (TP + FN) / (TP + TN + FP + FN), which mathematically corresponds to something closer to the complement of specificity rather than accuracy; is this a typographical error and, if so, were all reported accuracy values computed using the correct formula (TP + TN) / (TP + TN + FP + FN)?

9. For the crack-geometry quantification you rely on orthogonal skeletonization, region growing and Hough-based orientation estimation, but you do not report any comparison with ground-truth physical measurements (e.g., manual crack lengths and widths measured in the field or on calibration targets); what is the expected measurement error (bias and variance) of your pipeline and how sensitive is the RF safety prediction to these errors?

10. In the safety assessment results, the Random Forest achieves 87.0% accuracy and perfect recall for class S but shows noticeable confusion between MD and SD, and yet no confidence intervals or statistical tests are reported for the comparison with GB, SVM, MLP and DT; given the relatively small dataset and the practical importance of distinguishing minor from severe damage, how robust and practically acceptable are these performance numbers?

minor questions

1. Throughout the paper you use both "security" and "safety" to describe the assessment task (e.g., "structural security assessment" vs "structural safety assessment");

2. Some symbols and abbreviations are introduced without clear definitions the first time they appear (for example, the definition of in Eq. (13) and the use of RF, GB and DT before being expanded in the text);

3. There are several minor grammatical issues and slightly awkward phrases (for instance, "the learning rate batch size and the number of iterations were set to 0.001, 32, and 200" and "Datasets for security assessment");

**Do you want your identity to be public for this peer review?** For information about this choice, including consent withdrawal, please see our Privacy Policy

Reviewer #2: No

Reviewer #6: No

---

## [Author Response · Author response to Decision Letter 3]

27 Dec 2025

Please refer to the attached document "Response to Reviewers -6".

---

## [Decision Letter · Decision Letter 3]

11 Jan 2026

Structural Damage Detection and Safety Assessment Method Based on Machine Vision and Machine Learning

PONE-D-25-17025R3

Dear Dr. Le,

We’re pleased to inform you that your manuscript has been judged scientifically suitable for publication and will be formally accepted for publication once it meets all outstanding technical requirements.

Kind regards,

Peng Geng

Academic Editor

PLOS One

Additional Editor Comments (optional):

Reviewers' comments:

Reviewer's Responses to Questions

**Comments to the Author**

Reviewer #2: All comments have been addressed

Reviewer #6: All comments have been addressed

2. Is the manuscript technically sound, and do the data support the conclusions?

Reviewer #2: Yes

Reviewer #6: Yes

3. Has the statistical analysis been performed appropriately and rigorously?

Reviewer #2: Yes

Reviewer #6: Yes

4. Have the authors made all data underlying the findings in their manuscript fully available?

Reviewer #2: Yes

Reviewer #6: Yes

5. Is the manuscript presented in an intelligible fashion and written in standard English?

Reviewer #2: Yes

Reviewer #6: Yes

Reviewer #2: (No Response)

Reviewer #6: (No Response)

**Do you want your identity to be public for this peer review?** For information about this choice, including consent withdrawal, please see our Privacy Policy

Reviewer #2: No

Reviewer #6: No

---

## [Editor Report · Acceptance letter]

PONE-D-25-17025R3

PLOS One

Dear Dr. Le,

I'm pleased to inform you that your manuscript has been deemed suitable for publication in PLOS One. Congratulations! Your manuscript is now being handed over to our production team.

Kind regards,

on behalf of

Dr. Peng Geng

Academic Editor

PLOS One